# The long-term relationship between oil price changes and economic growth from the perspective of the resource curse: An empirical study from Yemen

Ebrahim Abbas Abdullah Abbas Amer[1], Zhang Xiuwu[1]*, Ebrahim Mohammed Ali Meyad[2], Ali. M. Meyad[3], A. K. M. Mohsin[4], Arifur Rahman[4]

1 Institute of Quantitative Economics and Statistics, Huaqiao University, Xiamen City, Fujian Province, China, 2 School of Economics and Management, Chang'an University, Xi'an, Shaanxi, China, 3 School of Economics, Sichuan University, Chengdu, Sichuan, China, 4 School of Economics, Lanzhou University, Lanzhou, Gansu, China

* Zxwxz717@hqu.edu.cn

**Data Availability Statement:** Data have been uploaded as Supporting information file.

## Abstract

A common conundrum discussed in economic research revolves around the fact that nations endowed with plentiful natural resources often exhibit a lower gross domestic product (GDP). This conundrum is commonly called the "resource curse", where most empirical studies about the effects primarily focused on developed economies. At the same time, limited data is available regarding a burgeoning oil-exporting nation like the Republic of Yemen. This research endeavor aims to investigate the relationship between oil price Changes and Yemen's economic growth. Utilizing annual data spanning from 1990 to 2019, the study employs the auto-regressive distributed lag (ARDL) model to establish the long-term connection between oil price volatility and economic growth over both short and long timeframes. This study's outcomes indicate that oil price Changes have a significant positive relationship with Yemen's economic growth in both the long and short run. Oil rents show a significant negative relationship with economic growth in both the long and short run. The results of GLM, RLS, and GMM robustness checks are consistent with our model results. Based on these findings, we suggest that Yemen should diversify its economy by investing in agriculture and tourism, and focus on human capital, education, and research and development. These steps could reduce the economy's dependence on oil and enhance sustainable economic growth. These empirical insights and suggestions are particularly useful for policymakers as they help build sound external and economic policies to sustain long-term economic growth.

## 1. Introduction

Dutch disease refers to a phenomenon where a country's booming resource sector, such as oil or minerals, can have adverse consequences on various sectors of the economy, especially

**Funding:** The author(s) received no specific funding for this work.

**Competing interests:** The authors have declared that no competing interests exist.

agriculture and manufacturing. This often occurs as a result of the appreciation of the nation's currency exchange rate, making its non-resource exports less competitive on the global market [1].

Three academic models can describe Dutch disease. The Economic Structure Model [2] focuses on the shifts in economic structure caused by a resource boom. This model suggests that the influx of revenue from the resource sector leads to increased spending, driving up demand for goods and services. This can lead to inflation and higher wages, making the non-resource sectors less competitive. As the resource sector grows, it attracts more labor and investment, diverting resources away from other sectors, which can lead to a decline in manufacturing and agriculture, as well as a loss of skills in those sectors. The Exchange Rate Channel Model [3] emphasizes the role of exchange rate movements in Dutch disease. A resource boom can lead to higher export revenues from the resource sector, causing the country's currency to appreciate, making non-resource exports more expensive for foreign buyers and reducing their competitiveness, potentially leading to a decline in non-resource exports, including manufactured goods and agricultural products, and harming non-resource sectors, potentially leading to job losses and reduced economic diversification [4]. The Spending Effect Model [5] highlights the impact of increased government revenue from the resource sector, often through taxes or royalties. The additional revenue can lead to increased government spending on public projects and services, which boosts demand for goods and labor, causing inflation and higher wages, affecting the overall cost structure of the economy. As wages rise, non-resource sectors may struggle to compete for labor, potentially leading to a shift of workers from these sectors to the booming resource sector, further exacerbating the decline in non-resource sectors [1].

Since the 1970s, economists have consistently noted the significant influence of crude oil price fluctuations on economic stability. For instance, Hamilton (2008) highlighted that [5], over the past few decades, nine out of ten U.S. recessions were preceded by notable increases in oil prices. Additionally, recent oil price surges have sparked concerns about potential economic slowdowns in developed countries. To illustrate, after nearly four years of price stability, Brent crude oil prices in Europe dropped from $100 per barrel in September 2014 to under $46 per barrel by January 2015, representing a reduction of over 50% within eleven months. This sharp decline is one of the largest seen in the past 30 years, similar to the significant drop in 1985–86.

As a result, there has been growing interest in examining how fluctuations in oil prices impact the economy. Many scholars have explored this topic, producing important studies on the subject [6–8]. The relationship between oil price changes and economic growth can vary significantly depending on whether a country is an oil exporter or importer [9]. According to the Energy Information Administration (EIA), global economic performance remains highly sensitive to changes in oil prices, highlighting the need to understand these connections. For that, this study investigates whether Changes in oil prices have a significant impact on the economic growth of an emerging, oil-dependent nation. Specifically, it contributes to the literature on the resource curse by focusing on Yemen, an oil-exporting economy that has not been extensively studied. By analyzing the effect of oil price Changes on Yemen's economic growth from 1990 to 2019, this research addresses a gap in the field. The study uses an auto-regressive distributed lag (ARDL) model to explore both the short-term and long-term relationships between oil price Changes and economic growth [10].

The economic development in the Republic of Yemen, much like many other developing nations, has been marked by significant imbalances, primarily due to an overreliance on natural resources, specifically oil and natural gas. The Yemeni economy is heavily dependent on oil production and exports, with oil generating approximately 70% of government revenue,

contributing 80–90% of total exports, and forming the bulk of foreign exchange reserves. Prior to the discovery of oil in 1985, agriculture and manufacturing were the dominant sectors, accounting for 24% and 14% of GDP, respectively (World Bank, 1989). However, since 1987, Yemen's economic structure has shifted dramatically, with the industrial sector, particularly oil and gas, and the services sector becoming more prominent contributors to GDP. In contrast, both the agricultural and manufacturing sectors have seen significant declines. This shift toward oil dependence has left the economy highly exposed to fluctuations in global oil prices, underscoring the argument for diversification to reduce vulnerability and promote economic stability [10]. See Figs 1 and 2, which show the path of GDP in Yemen and oil prices over the study period.

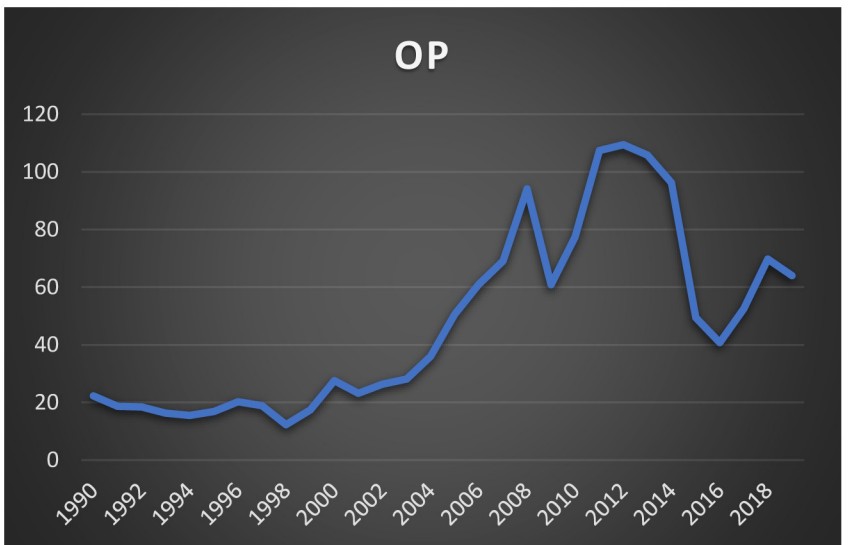

**Fig 1. The evolution of crude oil prices.**

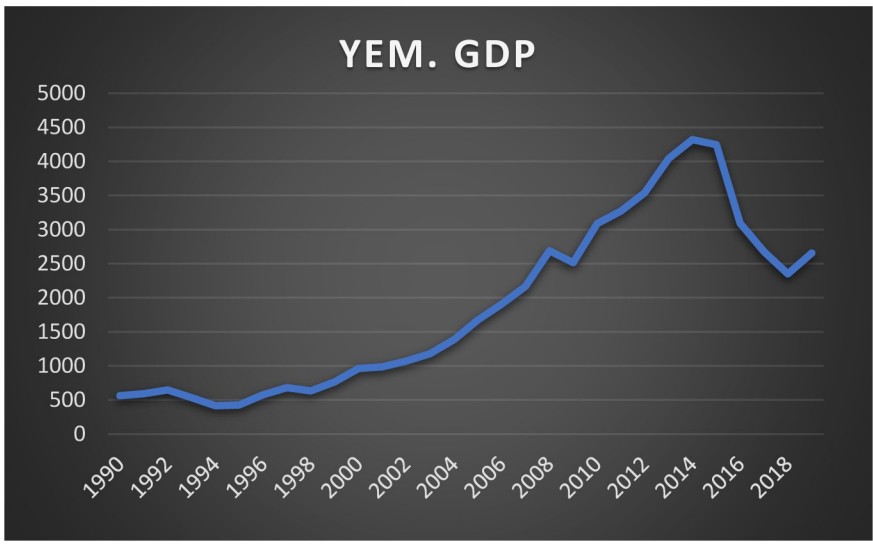

**Fig 2. The evolution of GDP.**

Moreover, the services sector has transitioned from its earlier role of primarily supporting agriculture and manufacturing to its present function of bolstering the oil industry, driven by the growing demand stemming from oil revenues [11]. Yemen, as a burgeoning oil-exporting nation, exemplifies the complexities of the relationship between oil prices and GDP due to its rapid development. Additionally, Yemen serves as a noteworthy case study, having engaged in financial sector activities since 1962, earlier than many other countries in the region. Initially, the financial sector experienced growth until the mid-1980s, but it began to decline during the oil boom. In 1995, the Yemeni government initiated a comprehensive reform program for the financial sector with good intentions [11]. However, despite these efforts, the country's economic performance continued to lag behind that of its regional counterparts, ranking as the lowest. Yemen exhibits numerous characteristics associated with the "oil curse," with its distinct symptoms prominently evident in the nation's economic landscape. These unique economic conditions and the persistent impact of oil dependency underscore the importance of our research in understanding and addressing the resource curse in Yemen [12].

In our paper, we describe Yemen as a "burgeoning oil-exporting nation" to highlight its relatively recent entry into the global oil market, starting significant exports in the late 1980s and developing its oil sector in the 1990s. This status is crucial as it allows us to examine the impact of oil price Changes on an economy at a different developmental stage compared to mature oil-exporting countries, adding diversity to the understanding of the resource curse. While pre-oil data for Yemen is scarce, our focus on the period from 1990 to 2019 captures the most impactful years of Yemen's oil-driven economic development. Our study fills a gap in existing literature by exploring these dynamics in a newer oil-exporting country, offering valuable insights for policymakers in similar contexts to design strategies that mitigate negative impacts and harness potential benefits.

The relationship between oil price Changes and GDP differs for countries at varying stages of oil production, and this difference is especially pronounced for nations that are just beginning to produce oil. Emerging oil producers often experience heightened economic volatility due to their reliance on oil as a major, if not sole, source of revenue. Unlike established producers with more diversified economies and robust institutions, early-stage producers typically lack the infrastructure and financial systems to buffer against oil price fluctuations. This makes their GDP more sensitive to global price Changes, as revenue from oil exports constitutes a larger proportion of their income. Furthermore, these nations are often grappling with underdeveloped governance structures, which may struggle to effectively manage oil revenues or invest in sustainable economic diversification, amplifying their vulnerability to changes in the oil market. Thus, the dynamics of oil price-GDP relationships for countries at the early stages of oil production may be fundamentally different, warranting focused analysis.

Our research paper provides a distinct contribution by focusing on the Republic of Yemen, a burgeoning oil-exporting nation that has not been extensively studied in the context of oil price Changes and economic growth. While previous studies as Victor & Ogbonna (2018) have established a positive impact of oil price Changes on economic growth, their study primarily concerns developed economies [13]. Our study fills a significant gap in the literature by investigating this relationship in Yemen, a developing country with different economic structures and dependencies. The study period from 1990 to 2019 captures significant economic events and oil price Changes that have uniquely impacted Yemen's economy. This research updates the literature with recent data and reflects the evolving economic conditions in Yemen, offering timely and relevant insights.

Yemen presents an important case study of an emerging oil producer. Like many nations entering oil production, Yemen relies heavily on oil revenues to fuel its economic growth, which makes it highly susceptible to global price shifts. However, Yemen's relatively low oil

reserves, combined with its protracted political instability, distinguish it from more typical emerging producers. Countries such as Angola or South Sudan, for instance, may have similar reliance on oil but differ in terms of political and infrastructural stability, as well as the scale of their oil reserves. Despite these differences, Yemen shares key challenges common to early-stage producers, such as the struggle to diversify the economy, manage oil revenues effectively, and develop institutions capable of harnessing the benefits of oil for broader development. While Yemen may not be entirely representative, its experiences offer valuable insights into the economic trajectories of countries facing the dual challenges of nascent oil production and unstable governance.

We employ the ARDL model to explore both the short-term and long-term effects of oil price volatility on Yemen's economic growth. This methodological approach, coupled with robustness checks using GLM, RLS, and GMM, ensures the reliability and validity of our findings [10]. These methodological advancements differentiate our study from previous research and contribute new insights into the economic dynamics of oil price Changes in a developing context. Additionally, our study offers practical policy recommendations for Yemen, emphasizing economic diversification, investment in agriculture and tourism, and a focus on human capital development, education, and research and development. These suggestions are crucial for reducing Yemen's dependence on oil and promoting sustainable economic growth. The policy implications derived from our findings are specifically tailored to address the unique economic challenges faced by Yemen, providing actionable insights for policymakers. These empirical insights hold implications for Yemen and broader discussions on sustainable economic growth and resource-dependent economies.

In the following sections of this paper, our focus has been directed towards the critical body of literature. The subsequent section provides an extensive elucidation of the data, variables, and methodology applied in this study. Following that, the fourth segment presents an analysis of the findings and engages in discussions pertinent to this research. To conclude, the fifth section encapsulates the key observations, impacts, and offers recommendations for policymakers in Yemen.

## 2. Literature review

### 2.1 Theoretical background

Subsequent to the groundbreaking work of [14], a considerable body of literature has emerged, delving into various transmission mechanisms that elucidate the impact of natural resources on economic growth. This line of research aims to explore whether it is feasible to mitigate the so-called natural resource curse by enhancing the quality of institutional frameworks. Additionally, scholars have probed into the nuances of the natural resource curse, evaluating its sensitivity to measurement techniques and the specific types of natural resources involved. In this context, Brunnschweiler & Bulte (2008) contribute significantly by distinguishing between two key aspects: resource dependence, which assesses the extent to which countries rely on natural resource exports, and resource abundance, which quantifies the wealth of a nation's natural resources [14]. Consequently, their findings fail to lend credence to the notion of the natural resource curse.

Researchers like (Bahar & Santos, 2018; Mehrara, 2008; Sala-i-Martin & Subramanian, 2008) have documented that the discovery of new oil reserves often leads to a real exchange rate appreciation and adverse effects on other export sectors of the economy [15, 16]. Meanwhile, a study by [17] indicated mixed outcomes, with some negative and some positive impacts. Nevertheless, natural resources can positively impact economic development, particularly by fostering the growth of the manufacturing sector in countries rich in such resources.

According to a study [18], about 40% of research papers analyzed the negative effects of resource abundance on economic growth, while another 40% found no clear impact, and the remaining 20% highlighted a positive correlation with economic growth. In contrast, Zallé (2019) supports the concept of the resource curse in African nations. He suggests that improving human capital and tackling corruption could allow these countries to turn the curse into a benefit [19, 20]. A separate analysis focusing on Algeria found a stable long-term relationship, though no significant short-term effects were observed. The results also indicated the dependence of the economy on hydrocarbons, which means that economic growth is subject to factors affecting oil prices.

In 2010, Özlale & Pekkurnaz (2010) conducted an analysis of the Turkish economy, exploring the connections between oil prices and various macroeconomic factors [21]. To do so, they utilized a structural vector autoregression model (SVAR) and established that Changes in oil prices were correlated with both a current account deficit and a reduction in economic growth. A comparable pattern was identified in China, as noted by Tang et al. (2011), where oil price fluctuations were shown to negatively affect both economic growth and investment [22]. Similarly, studies by Alley et al. (2014); and Moshiri & Banihashem (2012) expanded the analysis of oil price shocks and economic performance, focusing on countries such as the G-7, OPEC members, Russia, India, and China [20, 21]. Their findings revealed a negative relationship between oil prices and economic growth in oil-importing nations, while oil-exporting countries experienced a positive association [9, 23, 24]. Timilsina (2014) also broadened the scope by examining 25 economies, concluding that in developing nations, rising oil prices had a significantly detrimental effect on GDP [25]. This negative relationship was largely attributed to the heavy reliance of industries on oil as a key resource.

Moreover, the findings support the idea that higher oil prices contribute to the economic stability of oil-exporting nations. Ftiti et al. (2016) explored the link between oil prices and economic growth by analyzing monthly data from selected OPEC countries between 2000 and 2010 [8]. Their study highlighted that fluctuations in oil prices affect the relationship between oil and economic performance, especially during global economic cycles and the financial crisis in the OPEC region. Similarly, Shahbaz et al. (2017) examined data from 210 countries, emphasizing the significant impact of oil prices on both short-term and long-term growth [23]. Akinsola & Odhiambo (2020) as well as Artami & Hara (2018) explored the concept of asymmetry, indicating that the impact of oil price changes on economic growth can vary depending on whether prices rise or fall [22, 24]. Additionally, research by Ferrara et al. (2022), Su et al. (2020) and Kang et al. (2020) focused on the effects of oil price uncertainty on growth, suggesting that increased uncertainty may negatively influence economic performance [25–27]. These studies emphasize the importance of a detailed and context-specific analysis to better understand how oil price fluctuations affect economies through various complex channels.

## 2.2 The studies of natural resources and economic growth

The complex interplay between natural resources and economic growth continues to be a focal point in modern economic research. In the past, theories like the "Resource Curse" hypothesis postulated that nations abundant in natural resources might undergo diminished economic growth because of tendencies toward rent-seeking behaviors and a lack of investment in non-resource sectors [28]. However, more recent insights have brought nuance to this understanding. Drelichman & Voth (2022); and Lawer et al. (2017) challenged the universality of the curse, revealing that well-managed can mitigate its impact [29, 30]. Another relevant phenomenon, the "Dutch Disease," has been analyzed extensively, with many studies demonstrating

how the real exchange rate appreciation can impact economic diversification [31, 32]. Recent scholarship, exemplified by Chen & Lee (2014), has emphasized the role of policy frameworks and governance in shaping the outcomes of resource-rich economies, acknowledging the heterogeneity in resource-driven growth experiences [33]. Over the years, the role of natural resources in fostering the economic development of numerous nations has been substantial. According to research conducted by Erum & Hussain (2019), there is a notable correlation between the presence of natural resources and economic growth in countries that have embraced information and communication technology (ICT) more extensively [34].

Notably, the same research findings indicate that in economies with limited ICT diffusion, a negative correlation exists between natural resources and economic growth. However, it is essential to recognize that the effect of natural resources on economic growth is not always positive. The extent of their impact can depend on several other factors. For example, Raggl (2017) argues that natural resources may only promote economic growth when strong institutions and effective anti-corruption measures are in place [35]. Maximizing the benefits of natural resources requires improvements in institutional quality, which can, in turn, stimulate economic growth [36]. Empirical studies have shown that natural resources may hinder the economic development of resource-rich countries. For example, research has indicated that foreign direct investment (FDI) can positively impact economic growth in African nations, but only when certain conditions, such as adequate population and human capital levels, are met [37]. The relationship between natural resources, especially oil, and economic growth has been widely studied, particularly in Middle Eastern economies.

## 2.3 The studies of oil prices and economic growth

The relationship between oil prices and economic growth has been widely explored in academic research. For instance, Goel & Morey (1993) analyzed this connection within the U.S. economy, finding that rising oil prices typically hinder economic activity and growth [38]. Similarly, Hamilton (2008) studied data from OECD nations and observed that the effects of oil price fluctuations on economic growth can vary. Oil price shocks may either boost or constrain growth, depending on whether a nation primarily exports or imports oil [6].

Lardic & Mignon (2006) explored how oil prices relate to economic growth by employing asymmetric cointegration analysis. Their study identified a long-term relationship between the variables, concluding that higher oil prices typically dampen economic growth [39]. In the case of oil-exporting countries, Mehrara & Mohaghegh (2011) found that the relationship between oil prices and GDP was both non-linear and asymmetric, meaning the effect of oil prices on economic growth varied depending on the price levels [16]. Finally, Farzanegan & Markwardt (2009) analyzed the link between oil prices and several macroeconomic indicators in Iran, demonstrating that a positive oil price shock significantly boosted industrial production [40].

On the other hand, a drop in oil prices can negatively affect industrial output. Jayaraman & Choong (2009) conducted a study to examine how oil prices affect the economic growth of countries dependent on oil imports. Their research revealed that oil prices have a significant negative impact on economic growth, with a unidirectional causal relationship from oil prices to growth [41]. More recent research by scholars such as Sadorsky) 2012() 2014), Basher & Sadorsky (2016), and Nusair (2016), as well as Thorbecke (2019), has expanded upon this analysis, revealing diverse effects of oil price Changes on economic growth across different countries. These effects are influenced by varying degrees of oil dependence and distinct economic structures in these nations [42–46].

The existing research on the impact of oil price Changes on economic growth and its link to the resource curse provides no clear consensus. The nature of the relationship tends to differ based on whether a country is a net exporter or importer of oil. However, many studies that focus on developing countries suggest a negative relationship between oil prices and economic growth. The findings for these economies vary significantly, which can be explained by differences in factors like the choice of variables, model structures, monetary policies, and the unique economic features of each nation.

## 3. Data and methodology

### 3.1 Data description and sources

This study seeks to investigate the extent of the impact of oil price Changes on the growth of Yemen's economic condition from 1990 to 2019. Due to the difficulty obtaining data for the years preceding and following this period, this period has been adhered to. Following the example of all previous studies, we have used GDP as an indicator of economic growth, the dependent variable in our econometrics equation. As shown in Table 1, The World Bank, Statista, United Nations Statistics, Central Bank of Yemen and Arab Monetary Fund, and Statistical Year Book are the sources and references for each of the following variables (oil price, the Gross Domestic Product, government expenditure, exchange rate, Oil rents, Inflation rate, and the rate of unemployment) were collected.

In this paper, we define oil rents as the difference between the value of crude oil production at world prices and the total costs of production, representing a crucial component of national income for oil-exporting countries like Yemen. Oil rents are directly related to GDP as they constitute a significant portion of national revenue, influencing government spending and economic stability. Studying oil rents is important in the context of the resource curse theory, which suggests that countries with abundant natural resources often experience less economic growth [47]. Our study provides an empirical analysis of the relationship between oil rents and GDP in Yemen, a relatively recent oil-exporting country, offering fresh perspectives on how oil rents influence economic growth and informing policymakers in similar contexts about the potential benefits and challenges of relying on oil rents for economic development [48].

We included the exchange rate, inflation rate, government expenditure, and rate of unemployment as control variables in our model because they are key macroeconomic indicators that can influence GDP. While it is true that Changes in oil prices can affect these variables, their inclusion helps to isolate the specific impact of oil price Changes on GDP by accounting for these broader economic factors [48]. This approach is supported by econometric theory,

**Table 1. Data description and sources.**

| Variables | Symbol | Sources | The description |
|---|---|---|---|
| Gross Domestic Product | *GDP* | World Bank. | This variable represents a gross domestic product (in current U.S. dollars) as a globally recognized indicator of economic growth. |
| oil price | *OILP* | Statista. | This variable represents the Average annual OPEC crude oil price. |
| Oil rents | *OILR* | World Bank. | This variable represents the Oil rents (% of GDP). |
| government expenditure | *G.E.* | National Accounts Estimates of Main Aggregates and United Nations Statistics Division. | This variable represents the General government's final consumption expenditure. |
| exchange rate | *EXR* | Central Bank of Yemen and Arab Monetary Fund. | Concerning this variable, it represents the Domestic Currency Per U.S. Dollar (Period Average). |
| Inflation rate | *INF* | Statista. | The inflation rate during this study period. |
| The rate of unemployment | *UNE* | Statistical Year Book and UNITED NATIONS DEVELOPMENT PROGRAMME. | This variable represents the LU1 Unemployment rate for the Republic of Yemen during this study period. |

which suggests that including relevant controls can help to obtain more accurate estimates of the variable of interest (in this case, oil prices) on the dependent variable (GDP). Numerous empirical studies have demonstrated the importance of including control variables that can potentially mediate the relationship between the main explanatory variable and the dependent variable, enhancing the robustness of the results [10].

In order to investigate how Changes in oil prices and oil rents have influenced Yemen's economic growth from 1990 to 2019, we incorporated several control variables into our econometric model. These control variables, namely government expenditure, exchange rate, inflation rate, and the rate of unemployment, were included because they are known to have an impact on economic growth, which we are studying as the dependent variable. Oil price was the independent variable in our analysis. To enhance the efficiency and robustness of our experimental tests, we applied the natural logarithm transformation to all the variables. This transformation serves to improve the normal distribution of the variables, enhances data organization, and mitigates issues related to autocorrelation among the variables [49, 50]. Thus, the model and econometric equations that we will conduct time-series data experimental analyses on are:

$$GDP = f(OILP, EXR, GE, OILR, INF, UNE) \tag{1}$$

$$GDPt = \phi_0 + \phi_1 OILPt + \phi_2 EXRt + \phi_3 GEt + \phi_4 OILRt + \phi_5 INFt + \phi_6 UNEt + \varepsilon_t \tag{2}$$

After taking the natural logarithm of the variables, the equation is written as follows;

$$Ln\ GDPt = \phi_0 + \phi_1 \ln OILPt + \phi_2 \ln EXRt + \phi_3 \ln GEt + \phi_4 \ln OILRt + \phi_5 \ln INFt + \phi_6 \ln UNEt + \varepsilon_t \tag{3}$$

Where $\phi 1$, $\phi 2$, $\phi 3$, $\phi 4$, $\phi 5$, and $\phi 6$ are the coefficients of the independent and controlling variables [Oil price (OILP), government expenditure (G.E.), exchange rate (EXR), oil rents (OILR), inflation (INF)and the rate of unemployment (UNE)]. The "t" spans the years between 1990 and 2019. "ln" signifies the use of the natural logarithm, "$\phi 0$" stands for the intercept term, "$\phi$" represents the parameters, and "$\varepsilon$" is used to denote the error term.

As shown in (Table 2), Victor & Ogbonna (2018) have proven in their study that oil price Changes have a positive impact on economic growth, which means that when oil prices rise, people become more optimistic. Governments prefer to spend to fulfill recognized requirements, increasing the GDP rate [13]. That is, when oil revenues rise, the GDP rate also rises. This means that oil price Changes affect government spending, which determines the economy's development. Table 2 illustrates the associations between economic growth and various factors, including government expenditure, exchange rates, and the unemployment rate. Our research anticipates a positive correlation between government expenditure, oil rents, and exchange rates with regard to Yemen's economic growth. In contrast, unemployment and inflation are expected to negatively impact economic growth.

## 3.1 Methodology

**3.1.1 Descriptive statistics and correlation matrix.** We begin by conducting preliminary statistical tests to assess the characteristics of the variables used in the regression analysis. Key statistics include the minimum, maximum, mean, standard deviation, skewness, kurtosis, and Jarque-Bera values for each variable individually. Additionally, we examine the correlation matrix to understand the relationships between the variables under consideration. As well as to find out the extent of the interrelationship between the variables to each other [51].

**Table 2. The association between the dependent variable and independent variables in previous studies.**

| independent Variables | Researchers and date | Relationship to the dependent variable (GDP) |
|---|---|---|
| OILP | Zied Ftiti et al., 2016. Alley, Ibrahim et al., 2019 | The link between oil prices and economic growth in the context of OPEC nations, specifically Nigeria, demonstrates a mixed pattern. While some countries experience a positive correlation between oil prices and economic growth, others exhibit a negative relationship, where Changes in oil prices adversely affect their economic growth. |
| G.E. | Dash, Ranjan Kumar, and Chandan Sharma. "Government expenditure and economic growth: Evidence from India." The IUP Journal of Public Finance 6.3 (2008): 60–69 | The positive impact of economic growth. |
| EXR | Habib, M. M., Mileva, E., & Stracca, L. (2017). The real exchange rate and economic growth: Revisiting the case using external instruments. Journal of International Money and Finance, 73, 386–398 | They conducted a study to analyze how fluctuations in the real exchange rate affect the economic growth of a diverse panel of more than 150 countries. The findings validate this influence, but it appears to be most significant in the case of developing nations and countries with fixed exchange rate regimes. |
| OILR | Fuinhas, J. A., Marques, A. C., & Couto, A. P. (2015). Oil rents and economic growth in oil-producing countries: evidence from a macro panel. Economic Change and Restructuring, 48(3), 257–279. | Income derived from oil resources has a detrimental impact on economic growth, both in the immediate term and over the long term, indicating that it can be viewed as a burden rather than a benefit for economies. |
| INF | Barro, Robert J. "Inflation and economic growth." (1995), Gokal, V., & Hanif, S. (2004). | The negative impact of economic growth. |
| UNE | Calmfors, Lars, and Bertil Holmlund. (2000) | A higher growth rate can have both positive and negative unemployment effects. |

**3.1.2 Unit root test.** We employed the Augmented Dickey-Fuller (ADF) method, as introduced by Dickey & Fuller (1979) [52], and the Phillips and Perron (P.P.) method developed by Phillips & Perron (1988) to conduct a unit root test [53]. This test was performed to assess the stability of the variables within each series of our dataset and to investigate the time-series characteristics of each variable, ultimately determining its level of integration. In this test, the null hypothesis assumes the existence of a unit root for each time series, while the alternative hypothesis posits the absence of a unit root in each time series [54].

As we know, the equations of the Dickey-Fuller test for the unit root can be written as follows:

$$\Delta y_t = c + a y_{t-1} + \sum_{j=1}^{k} d_j \Delta y_{t-1} + \varepsilon_t. \tag{4}$$

$$\Delta y_t = c + a y_{t-1} + \beta_t + \sum_{j=1}^{k} d_j \Delta y_{t-1} + \varepsilon_t. \tag{5}$$

**3.1.3 ARDL model.** This article utilizes the auto-regressive distributed lag (ARDL) bound testing method of cointegration, as proposed by (Pesaran et al., 2001). The ARDL approach is employed to establish the enduring and immediate connections between GDP and Changes in oil prices, while also examining the links between GDP and other independent variables. Recent research indicates that the ARDL model surpasses the Engle and Granger technique (1987) and the Johansen approach (1988) when it comes to estimating cointegration relationships due to its enhanced reliability and applicability, irrespective of whether the underlying regressors are I(0) or I(1). Additionally, this method excels in handling small sample sizes and can simultaneously assess the short- and long-term impacts of independent variables on the dependent variable [55]. Finally, all variables are endogenous, eliminating the endogeneity issues plaguing the Engle-Granger approach.

To estimate the impact of the short- and long-run explanatory variables on economic growth, the ARDL model is represented according to the following equation:

$$
\begin{aligned}
\Delta GDP_t = {} & \beta_0 + \sum_{i=2}^{q} \beta_{1i} \Delta GDP_{t-i} + \sum_{i=2}^{q} \beta_{2i} \Delta OILP_{t-i} + \sum_{i=2}^{q} \beta_{3i} \Delta GE_{t-i} + \\
& \sum_{i=2}^{q} \beta_{4i} \Delta EXR_{t-i} + \sum_{i=2}^{q} \beta_{5i} \Delta UNE_{t-i} + \sum_{i=2}^{q} \beta_{6i} \Delta INF_{t-i} + \\
& \sum_{i=2}^{q} \beta_{7i} \Delta OILR_{t-i} + \alpha_1 GDP_{t-1} + \alpha_2 OILP_{t-1} + \alpha_3 GE_{t-1} + \alpha_4 EXR_{t-1} + \\
& \alpha_5 UNE_{t-1} + \alpha_6 INF_{t-1} + \alpha_7 OILR_{t-1} + \varepsilon_t
\end{aligned} \tag{6}
$$

Where $\Delta$ represents the First difference coefficient. $\beta_0$ represents the intercept term, $\beta_1$, $\beta_2$, $\beta_3$, $\beta_4$, $\beta_5$, $\beta_6$, and $\beta_7$ The parameters are indicative of short-term effects, which capture the immediate effects of changes in the explanatory variables on the dependent variable, $\alpha_1$, $\alpha_2$, $\alpha_3$, $\alpha_4$, $\alpha_5$, $\alpha_6$, and $\alpha_7$ Parameter indicative of the long-term dynamics of the model, there is only one alpha per variable because the long-run coefficients reflect the cumulative, steady-state impact of the explanatory variables on the dependent variable once equilibrium has been reached. Finally, $\varepsilon$ is the error term.

The reason the long-run coefficients are in logs of the variables while the short-run coefficients are expressed in changes in the log of the variables (as seen in Eq 6 and Table 8) is due to the nature of the ARDL model. In the short run, we are interested in how the changes (differentials) of the independent variables affect GDP. In the long run, however, the model estimates the relationship between the levels of the variables in their logged forms, reflecting how a proportional change in one variable affects the other after all adjustments have taken place.

**3.2.4 Toda-Yamamoto test.** The Toda-Yamamoto causality test, developed by Toda and Yamamoto (1995), is an extension of the traditional Granger causality test. It is particularly useful in dealing with time series data that may be non-stationary or integrated of different orders. Unlike standard causality tests that require pre-testing for stationarity and cointegration, the Toda-Yamamoto approach circumvents these issues by estimating an augmented Vector Autoregressive (VAR) model [56]. This method adds additional lags to account for potential non-stationarity, allowing for the testing of causality without the risk of misidentifying the integration order of the variables [57].

The test proceeds by estimating the VAR model with additional lags, after which a Wald test is conducted on the coefficients of the lagged explanatory variables. This helps determine whether the lagged values of one variable (e.g., oil prices) have predictive power over the other variable (e.g., GDP). By doing so, the Toda-Yamamoto test offers a robust way to infer directional causality between variables even when there is uncertainty about their stationarity. This approach has been widely used in economic research due to its flexibility and reliability in dealing with mixed-order integration time series [58, 59].

### 3.3 Robustness examination tests

Analyzing the robustness of examination tests is a sound epistemic approach when we acknowledge the potential fallibility of our assumptions and inferences. This is particularly relevant in economics, which distinguishes itself from certain fields of physics due to the inherently diverse, dynamic, and open nature of the systems it investigates, as pointed out by [60]. To validate the relationships among all the variables in our study, we applied robustness examination tests, specifically employing the Generalized Linear Model (GLM), Robust Least

Squares (RLS), and Generalized Method of Moments (GMM) methods. These tests were conducted to ensure the reliability of the experimental findings, as detailed by [61].

# 4. Results and discussion

## 4.1 Descriptive statistics

Table 3 shows that oil rent is the most volatile, followed by the annual inflation rate. Real exchange rate, while GDP has less deviation, government spending and oil price are less volatile, and the unemployment rate is considered the least variable compared to all other variables. Through the results of the same Table, it is clear that all the study variables are distributed normally because (prob. jarque bera >5%) except the real exchange rate and oil rent variables.

## 3.5 Correlation matrix results

Table 4 presents the outcomes of the correlation matrix examination, highlighting notable correlations between the GDP, our dependent variable, and various explanatory factors like government expenditure, exchange rate, oil price, and unemployment rate. These correlations are notably strong, with coefficients surpassing the 0.5 threshold. Furthermore, the correlations between oil prices and the unemployment rate with exchange rate and government expenditure are particularly noteworthy. However, it's important to note that the causal analysis we plan to undertake in the subsequent tests will provide a more definitive assessment of these relationships.

**Table 3. Results for descriptive statistics.**

|  | LGDP | LOILP | LOILR | LEXR | LGE | LINF | LUNE |
|---|---|---|---|---|---|---|---|
| Mean | 23.37528 | 3.637190 | 2.860866 | 4.918213 | 21.52710 | 2.806932 | 2.429172 |
| Median | 23.44732 | 3.646304 | 3.270354 | 5.233201 | 21.41116 | 2.491463 | 2.494031 |
| Maximum | 24.48926 | 4.695468 | 3.745153 | 5.521461 | 22.52230 | 4.266616 | 2.602690 |
| Minimum | 22.15055 | 2.507972 | -0.369471 | 2.633327 | 20.49903 | 1.302913 | 2.104134 |
| Std. Dev. | 0.767370 | 0.694394 | 1.067819 | 0.768426 | 0.601980 | 0.779995 | 0.173439 |
| Skewness | -0.096601 | 0.072120 | -1.645974 | -1.759425 | 0.029744 | 0.316719 | -0.891970 |
| Kurtosis | 1.560063 | 1.588283 | 4.818665 | 4.938349 | 1.636779 | 2.273561 | 2.302508 |
| Jarque-Bera | 2.638433 | 2.517187 | 17.68058 | 20.17438 | 2.327388 | 1.161198 | 4.586172 |
| Probability | 0.267345 | 0.284053 | 0.000145 | 0.000042 | 0.312330 | 0.559563 | 0.100954 |
| Sum | 701.2584 | 109.1157 | 85.82599 | 147.5464 | 645.8129 | 84.20795 | 72.87515 |
| Sum Sq. Dev. | 17.07682 | 13.98331 | 33.06689 | 17.12388 | 10.50901 | 17.64338 | 0.872347 |
| Observations | 30 | 30 | 30 | 30 | 30 | 30 | 30 |

**Table 4. Results for the correlation matrix.**

|  | ln GDP | ln OP | ln OILR | ln EXR | ln GE | ln INF | ln UNE |
|---|---|---|---|---|---|---|---|
| **ln GDP** | 1.000000 |  |  |  |  |  |  |
| **ln OP** | 0.931659 | 1.000000 |  |  |  |  |  |
| **ln OILR** | -0.527337 | -0.296905 | 1.000000 |  |  |  |  |
| **ln EXR** | 0.714471 | 0.621873 | -0.275130 | 1.000000 |  |  |  |
| **ln GE** | 0.951467 | 0.945977 | -0.438399 | 0.534902 | 1.000000 |  |  |
| **ln INF** | -0.464612 | -0.404137 | -0.107923 | -0.609978 | -0.398311 | 1.000000 |  |
| **ln UNE** | 0.894941 | 0.782536 | -0.424630 | 0.910807 | 0.760971 | -0.646956 | 1.000000 |

**Table 5. Results for unit root test.**

| Variable | (ADF) Test | | PP Test | |
|---|---|---|---|---|
| | *Level* | *1st difference* | *Level* | *1st difference* |
| ln *GDP* | -1.0437 | -3.3487** | -1.0650 | -3.3210** |
| ln *OILP* | -1.0050 | -4.7193*** | -1.0064 | -4.7028*** |
| ln *OILR* | -1.6896 | -4.5072*** | -1.2860 | -3.44133** |
| ln *EXR* | -8.0824*** | _______ | -20.653*** | _______ |
| ln *GE* | -0.7623 | -4.4440*** | -0.8685 | -4.4455*** |
| ln *INF* | -2.6818* | _______ | 2.6818* | _______ |
| ln *UNE* | -2.4173 | -2.9075*** | -2.0533 | -2.9103* |

Note:

***, **, and * show significance at the 1%, 5%, and 10% levels, respectively.

## 3.6 Unit root test result

In Table 5, we assess the stability of the variables using unit root tests, specifically the Augmented Dickey-Fuller (ADF) and Phillips-Perron (PP) tests. The results in Table 1 show that the exchange rate and inflation variables are stationary at their levels, allowing us to reject the null hypothesis of non-stationarity. Conversely, for the other variables, the probability values suggest that we cannot reject the null hypothesis, indicating they are non-stationary at their levels. However, after considering the first difference, the null hypothesis is rejected, confirming stationarity. In summary, all variables become stationary at their first differences, except for the exchange rate and inflation, which are stationary at their levels. This indicates that the variables have mixed orders of integration, specifically I(0) and I(1), justifying the use of the Autoregressive Distributed Lag (ARDL) model for our analysis.

## 3.7 ARDL test result

Upon conducting an investigation into the time series characteristics of all the variables, the ARDL methodology was employed to assess the potential long-term equilibrium relationship. It's important to note that the outcome of this test can be influenced by the number of lags considered. In view of the limited amount of data available for this study, we opted to incorporate lags of up to two years on the first difference of each variable. The selection of the optimal lag length for each variable was based on the AIC criterion. The AIC criterion led to the recommended model: ARDL (1, 1, 1, 2, 1, 2, 2). The findings of the ARDL bound test for cointegration can be found in Table 6.

In Table 6, we can observe that the computed F-statistic surpasses the critical value upper bound, indicating compelling evidence of a sustained association among economic growth, oil

**Table 6. Results for ARDL bound test.**

| Test Statistic | Value | k |
|---|---|---|
| F-statistic | 15.77319 | 6 |
| **Critical Value Bounds** | | |
| Significance | I0 Bound | I1 Bound |
| **10%** | 1.99 | 2.94 |
| **5%** | 2.27 | 3.28 |
| **2.5%** | 2.55 | 3.61 |
| **1%** | 2.88 | 3.99 |

**Table 7. Results for ARDL model stability diagnostic test.**

| Hypotheses | tests | (prob) |
|---|---|---|
| Autocorrelation | Breusch-Godfrey | 0.25 |
| Heteroskedasticity | Breusch-pagangodfrey | 0.88 |
| | Arch-test | 0.89 |
| Normality | Jarque-bera | 0.13 |
| Specification | Ramsey(fisher) | 0.16 |

price, oil rents, government expenditure, exchange rate, inflation, and unemployment in Yemen. The results shown in Table 7 show that there is no autocorrelation of errors, no Heteroskedasticity of errors, and no normal distribution of errors. The estimated model (1, 1, 1, 2, 1, 2, 2) is statistically acceptable, as the null hypothesis is accepted for all tests, as it explains about 98% of the dynamics of GDP during the period 1990–2019.

There is evidence of cointegration among the variables, signifying a long-term connection. To derive the long-term coefficients, we divide the estimated coefficient of the lagged independent variable by that of the lagged dependent variable, and then apply a negative sign to the result. In contrast, short-term coefficients are based on the estimated coefficients of the variables in their first-difference form.

In the short run, the connection between Changes in oil prices and Yemen's economic growth demonstrates a noteworthy positive correlation. More precisely, an upswing in short-term oil price variations is linked to a marked upturn in economic growth. This suggests that the economy tends to expand in response to short-term surges in oil prices. Conversely, Short-term changes in oil revenue are associated with a pronounced negative correlation with economic growth. A decrease in oil revenue during this period is found to be associated with economic growth, highlighting the economy's dependence on oil-generated income. While government spending registers a minor decline, it shows a significant association with economic growth at a 0.05 significance level. Similarly, a decline in the short-term exchange rate is observed to have a highly significant adverse relationship with economic growth. Furthermore, elevated short-term inflation rates demonstrate a significant negative relationship with economic growth, attributed to inflation's detrimental effects on purchasing power and economic stability. Interestingly, the relationship between short-term unemployment and economic growth is marginally significant. An increase in unemployment during this period may marginally bolster economic growth, revealing intricate dynamics at play.

About the enduring connection between economic growth and various factors, the findings disclosed in Table 8 suggest a noteworthy positive correlation between Changes in crude oil prices and economic growth. Consequently, a 1% rise in oil prices is associated with a 0.6666% increase in economic growth, with statistical significance at the 1% level. This alignment with economic theories underscores the relationship between international oil price changes and economic performance in oil-exporting nations, both in statistical and economic terms. This indicates that the Yemeni economy is a rentier economy par excellence. Therefore, the Yemeni economy is linked to the hydrocarbon sector, which affects it by the most important events that occur at the same level in case of increase or decrease.) This result is consistent with many prior studies that demonstrated the positive connection between economic growth and oil price Changes, such as [9, 16, 45, 46, 62–64].

The test results presented in Table 8 clearly indicate a noteworthy adverse correlation between oil rents and economic growth. In other words, when oil rents increase by 1%, economic growth decreases by 0.1061% at a 1% significance level, illustrating what is commonly

**Table 8. Results for ARDL short and long-run model.**

| Short Run | | |
|---|---|---|
| **Variable** | Coefficient | t-Statistic |
| *ΔlnOILP* | 0.6826*** | 20.5578 |
| *ΔlnOILR* | -0.2654*** | -16.7358 |
| *ΔlnGE* | -0.1145** | -2.9487 |
| *ΔlnEXR* | -0.3308*** | -7.4712 |
| *ΔlnINF* | -0.0894*** | -9.0319 |
| *ΔlnUNE* | 0.4974* | 1.8526 |
| *CointEq(-1)* | -0.8164*** | -14.3696 |
| Long Run | | |
| **Variable** | Coefficient | t-Statistic |
| *lnOILP* | 0.6666*** | 5.2001 |
| *lnOILR* | -0.1061*** | -3.7225 |
| *lnGE* | 0.0975 | 0.6444 |
| *lnEXR* | -0.1200 | -1.2968 |
| *lnINF* | -0.2085*** | -3.3915 |
| *lnUNE* | 0.8499 | 1.6174 |
| *C* | 18.3489*** | 5.8615 |
| **R- Squared** | **0.98** | |
| **S.E of regression** | **0.032** | |

Note:

***, **, and * show significance at the 1%, 5%, and 10% levels, respectively.

referred to as the "resource curse." Oil rents, in this context, represent the disparity between the value of crude oil production at regional prices and the total production costs, with a notable link to oil prices. It's important to note that the resource curse isn't solely attributed to the abundance of resources but also to the volatility in their prices. Contrary to classical theories positing that a wealth of natural resources benefits economic growth, our findings align with Sachs and Warner (1995), who empirically demonstrate that resources can indeed be detrimental to an economy. This outcome is constant with many previous works, such as [14, 65–67].

Government expenditure was found to have a significant negative association with economic growth. Therefore, an increase in government expenditure by 1% leads to a decrease in economic growth by 0.1145% in the short term at 5%, a significance level. In the long run, it shows an insignificant positive relationship with economic growth. Thus, the government expenditure on economic growth ought to have a significant positive coefficient [68]. Since it does not, this implies that Less Developed Country government expenditure is inefficient [69, 70]. That is, government spending was on unproductive economic activities and non-developmental projects, such as (investment, infrastructure, education, capital spending). The Yemeni government spent the most on current expenditures and spending on the security and military sector due to the security turmoil in Yemen during the study period. Also, these outcomes are in line with our expectations and consistent with many studies such as [68–70].

As anticipated, the empirical analysis revealed a statistically insignificant and adverse relationship between the real exchange rate and economic growth. Consequently, an appreciating real exchange rate is associated with a reduction in long-term economic growth. This relationship can be attributed to the significant role of the oil sector in the economy, which not only

accounts for approximately 70% of government revenues but also contributes a substantial share of exports (around 80–90%) and the bulk of foreign exchange reserves, primarily denominated in US dollars. This interdependence further highlights the link between oil price changes and the economy. Also, these outcomes are consistent with some studies, such as [71, 72].

The connection between the inflation rate and economic growth is notably adverse. Consequently, a 1% rise in the inflation rate results in a long-term economic growth decline of 0.2085% at a 1% significance level. Research findings in this area consistently indicate that this negative association is more pronounced in countries struggling to uphold price stability during periods of high inflation [72–74]. Finally, we found that the unemployment rate has an insignificant positive relationship with economic growth. Hence, when there is a shift in the inflation rate, it typically results in a long-term increase in economic growth. Consequently, we can infer that there exists a minor yet positive correlation between the unemployment rate and economic growth. It's important to note that economic growth doesn't directly decrease unemployment but does so indirectly by creating more job opportunities within the economy. On the other hand, economic growth rates may be due mostly to growth in sectors that employ a few workers due to their high capital densities. This happened in Yemen's oil and gas sector, which has doubled its production capacities several times but employed small workers.

## 3.8 Robustness examination test result

To confirm the relationships between the variables in this study, we applied multiple robustness checks, such as the Generalized Linear Model (GLM), Robust Least Squares (RLS), and Generalized Method of Moments (GMM). These methods were employed to guarantee the consistency and reliability of the study's results.

Table 9 presents a clear demonstration of the alignment between the outcomes obtained through the Generalized Linear Model, Robust Least Squares, and Generalized Method of Moments approaches and the models selected for this study, specifically the Autoregressive Distributed Lag (ARDL) model. The findings from GLM, RLS, and GMM methods consistently indicate a significant positive correlation between each of the oil prices and economic growth, while also revealing a significant negative relationship between oil rents and economic growth. It's noteworthy that the remaining variables yield results that are in harmony with all the models, reinforcing the notion that the results correspond to our chosen ARDL model, whether in the long or short term. This contributes to the robustness and reliability of the findings.

**Table 9. Results for RLS, GLM, and GMM models.**

| variables | RLS method | | GLM method | | GMM method | |
|---|---|---|---|---|---|---|
| | Coefficient | z-Statistic | Coefficient | z-Statistic | Coefficient | z-Statistic |
| *LOILP* | 0.3710*** | 3.6948 | 0.274** | 2.3068 | 0.4021*** | 3.7386 |
| *LOILR* | -0.1268*** | -4.8821 | -0.1268*** | -3.1372 | -0.1205*** | -3.9455 |
| *LGE* | 0.4239*** | 3.5448 | 0.5089*** | 3.5982 | 0.3936*** | 3.2593 |
| *LEXR* | 0.0331 | 0.4852 | 0.0002 | 0.0022 | 0.0294 | 0.4508 |
| *LINF* | -0.0089 | -0.2398 | 0.0003 | 0.0072 | -0.0066 | -0.2624 |
| *LUNE* | 1.2574*** | 2.8954 | 1.5049*** | 2.9296 | 1.2580*** | 3.3013 |

Note:

***, **, and * show significance at the 1%, 5%, and 10% levels, respectively.

**Table 10. Results for Toda Yamamoto causality test.**

| Variables | Variables | | | | | | |
|---|---|---|---|---|---|---|---|
| | **LGDP** | **LOILP** | **LOILR** | **LGE** | **LEXR** | **LINF** | **LUNE** |
| LGDP | _ _ | 2.016 (0.569) | 1.074 (0.783) | 1.214 (0.750) | 3.619 (0.306) | 0.512 (0.916) | 5.250 (0.154) |
| LOILP | 1.938 (0.586) | _ _ | 1.206 (0.751) | 4.624 (0.202) | 3.932 (0.269) | 3.342 (0.342) | 1.884 (0.597) |
| LOILR | 10.942 (0.012) | 5.913 (0.116) | _ _ | 6.361 (0.095) | 6.254 (0.010) | 7.314 (0.062) | 4.006 (0.261) |
| LGE | 0.093 (0.993) | 6.854 (0.077) | 7.002 (0.072) | _ _ | 5.009 (0.171) | 2.028 (0.567) | 0.458 (0.928) |
| LEXR | 1.938 (0.585) | 1.653 (0.648) | 1.115 (0.773) | 0.725 (0.867) | _ _ | 3.843 (0.279) | 3.691 (0.297) |
| LINF | 10.731 (0.013) | 3.841 (0.279) | 3.484 (0.323) | 10.851 (0.013) | 3.955 (0.266) | _ _ | 7.012 (0.072) |
| LUNE | 10.191 (0.0170) | 6.081 (0.108) | 7.186 (0.066) | 11.538 (0.009) | 34.001 (0.000) | 11.985 (0.017) | _ _ |

### 3.9 Toda Yamamoto causality test result

Table 10 presents the findings of this analysis, which reveal a one-way causal relationship where oil rents, the real exchange rate, and the inflation rate influence economic growth. In contrast, a reciprocal causal relationship exists between oil rents and government spending. Additionally, government expenditure is shown to have a one-way causal effect on both oil prices and economic growth. The analysis also identifies a bidirectional causal link between the inflation rate and the unemployment rate. Moreover, a one-way causal relationship is observed where the unemployment rate affects economic growth, oil rents, government expenditure, and the real exchange rate.

## 5. Conclusion and policy implications

Changes in oil prices in global markets are expressed as a result of economic conditions in supply and demand and international political events. This fluctuation in oil prices presents a list of threats to the exporting countries that depend on it mainly for the growth of their economies and consider it a curse, as in the event of a decline in oil prices, the total exports of these countries decline and thus paralysis in other economic sectors, as it is an indispensable financial resource for these sectors, but in the event of an increase in its prices and an increase in oil revenues, the damage may be more than the previous case, as this rise is accompanied by a significant increase in the revenues and incomes of these countries, which hurts the growth of other economic sectors, what is known as the "curse of the sources." As long as Changes continue to decrease and increase in oil prices, these economies in Yemen or oil-dependent countries are hostages to developments in the oil market.

This study utilizes time-series data analysis techniques to explore the impact of fluctuations in oil prices on the Gross Domestic Product (GDP), a key indicator of economic growth in the Republic of Yemen, covering the period from 1990 to 2019. To evaluate the stationarity of our dataset, we conducted the Augmented Dickey-Fuller (ADF) and Phillips-Perron (PP) unit root tests, which confirmed that all variables demonstrated stationary characteristics with different trends. To analyze the effects of oil price fluctuations on both the short- and long-term growth of the economy, we employed the Auto-Regressive Distributed Lag (ARDL) model. The results obtained from the Generalized Linear Model, Robust Least Squares, and Generalized Method

of Moments methods consistently corroborate the findings derived from the ARDL model used in this research.

In the short run, the association between Changes in oil prices and economic growth in Yemen reveals a significant positive association. Specifically, a rise in short-term oil price Changes is linked to a notable rise in economic growth. This suggests that the economy tends to experience growth in response to short-term spikes in oil prices. Conversely, short-term shifts in oil rents are significantly negatively associated with economic growth. A decrease in oil rents during this period is connected with economic growth, underlining the economy's reliance on oil revenue. Despite a slight decrease, government spending is significantly associated with economic growth at a 0.05 significance level.

Likewise, a reduction in the short-term exchange rate is observed to exert a substantial adverse impact on economic growth. Moreover, elevated short-term inflation rates are found to have a noteworthy detrimental association with economic growth, primarily stemming from the adverse consequences of inflation on both purchasing power and economic stability. Interestingly, the relationship between short-term unemployment and economic growth is marginally significant. An increase in unemployment during this period might slightly bolster economic growth, indicating complex dynamics at play.

Turning to the long-term perspective, the connection between economic growth and oil price Changes reveals a substantial and positive correlation. Sustained increases in oil prices over the long term are linked to significant economic growth in Yemen. This underscores the potential benefits of stable oil prices for long-term economic development. In contrast, variations in oil rents over the long term display a significant negative relationship with economic growth. This emphasizes that Changes in oil revenue can hinder economic growth in an extended period.

Interestingly, Government spending is not statistically significantly associated with long-term economic growth. This suggests that other factors might have a more prominent role in shaping economic growth trends over the long term. Likewise, exchange rate changes in the long term are not found to be significantly associated with economic growth, suggesting a nuanced interplay of multiple factors. Long-term inflation rates, however, exhibit a highly significant negative relationship with economic growth. This highlights the imperative of maintaining low and stable inflation for sustained economic growth. Lastly, interpreting the long-term relationship between unemployment and economic growth is challenging due to unusual p-value values, implying a need for further scrutiny and validation of this result.

The outcomes of this study offer several important implications for policy formulation and decision-making in Yemen. Given the significant positive relationship between short-term Changes in oil prices and economic growth, policymakers should consider strategies that harness the potential benefits of oil price volatility. Implementing mechanisms to leverage periods of elevated oil prices could allow the country to channel increased revenue towards targeted development projects, stimulating economic growth during these phases. The long-term perspective underscores the importance of stable oil prices for Yemen's sustained economic growth. Building on the significant positive relationship identified between long-term oil price stability and economic growth, policymakers should advocate for long-term contracts and agreements that provide stability in oil pricing. This could involve collaborations with international partners to ensure a predictable revenue stream, offering a foundation for consistent economic development efforts.

The relationship between oil prices and GDP in Yemen contrasts with the patterns observed in more established oil-producing countries. In long-standing oil economies like Saudi Arabia or Norway, the impact of oil price Changes is often mitigated by diversified economies, established sovereign wealth funds, and institutional mechanisms for managing oil revenue. These

countries also tend to have more stable governance, which helps cushion against the negative effects of price volatility. By contrast, Yemen, as an early-stage oil producer, is more vulnerable to external shocks due to its heavy reliance on oil income and the lack of such buffers. In comparing our findings to the existing literature on established oil producers, we note that Yemen's experience highlights the precarious nature of early oil production phases. The heightened sensitivity of GDP to oil price Changes underscores the need for more nuanced policy approaches tailored to the specific vulnerabilities of emerging oil economies, distinguishing them from their more developed counterparts.

Additionally, acknowledging the detrimental impact of short-term Changes in oil rents on economic growth calls for the establishment of mechanisms that buffer the economy from rapid shifts in oil revenue. Diversification efforts, such as developing non-oil sectors and strengthening domestic industries, could provide the resilience needed to mitigate the negative effects of oil rent volatility. The negative association between long-term changes in oil rents and economic growth highlights the need for a comprehensive national strategy that addresses oil revenue volatility. This includes diversification efforts that promote the growth of non-oil sectors, such as agriculture, manufacturing, and services. By reducing the economy's dependence on oil revenue, Yemen can enhance its economic resilience and mitigate the adverse effects of fluctuating oil rents.

Recognizing the marginally significant association between short-term unemployment and economic growth offers an avenue for labor market policies. Policymakers could explore initiatives providing training and skill development opportunities during increased unemployment. By enhancing the employability of the workforce, the economy could benefit from a more adaptable and skilled labor pool, potentially fostering economic growth even during challenging times. The substantial negative relationship between long-term inflation rates and economic growth necessitates prudent monetary policies to maintain low and stable inflation. Policymakers should work towards implementing measures that monitor and control inflationary pressures, such as effective monetary policy tools and prudent fiscal management. Ensuring price stability can create an environment conducive to economic growth, where consumers and businesses can confidently plan for the future.

## 5.1 Limitations of the study

The study has several limitations that may affect the generalizability. First, the study is focus on a single country, Yemen. Second, annual data spanning from 1990 to 2019, the study employs the auto-regressive distributed lag (ARDL) model to establish the connection between oil price volatility and economic growth over both short and long timeframes. Unfortunately, the data after 2019 is not available. Finally, the study is an empirical study. A mixed method analysis or using big data may discover the phenomenon better.

## 5.2 Future research direction

Future research may collect annual data spanning from 2020 to 2024 and employs the auto-regressive distributed lag (ARDL) model to establish the connection between oil price volatility and economic growth over both short and long timeframes with a comparison of the existing study. Furthermore, big data analytics may discover useful findings to ensure more generalizability. Researchers may compare the result of the study with similar countries and conduct a multi-country analysis on the similar aspect. While this paper examined the long-run relationship between price Changes and economic growth, we recommend that future studies investigate the more interesting topic of how variability in oil prices affects GDP in an oil-exporting country.

## Supporting information

**S1 File.**
(XLSX)

## Author Contributions

**Conceptualization:** Zhang Xiuwu.

**Data curation:** Ebrahim Mohammed Ali Meyad, Ali. M. Meyad, A. K. M. Mohsin.

**Formal analysis:** Ebrahim Abbas Abdullah Abbas Amer.

**Methodology:** Ebrahim Abbas Abdullah Abbas Amer.

**Writing – original draft:** Ebrahim Abbas Abdullah Abbas Amer.

**Writing – review & editing:** Zhang Xiuwu, Arifur Rahman.

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
