## [Decision Letter · Decision Letter 0]

17 Mar 2024

PONE-D-23-43590The relationship between oil price fluctuations and economic growth from the perspective of the resource curse: An empirical study from YemenPLOS ONE

Dear Dr. AMER,

Thank you for submitting your manuscript to PLOS ONE. After careful consideration, we feel that it has merit but does not fully meet PLOS ONE’s publication criteria as it currently stands. Therefore, we invite you to submit a revised version of the manuscript that addresses the points raised during the review process.

**ACADEMIC EDITOR: The reviewers have serious concern on your manuscript so improve the manuscript in light of given comments.**

We look forward to receiving your revised manuscript.

Kind regards,

Ghulam Rasool Madni, Ph.D

Academic Editor

PLOS ONE

Journal Requirements:

3. In the online submission form, you indicated that data will be made available on request. 

4. Please ensure that you refer to Figure 1 and 2 in your text as, if accepted, production will need this reference to link the reader to the figure.

Reviewers' comments:

Reviewer's Responses to Questions

**Comments to the Author**

1. Is the manuscript technically sound, and do the data support the conclusions?

Reviewer #1: No

Reviewer #2: Yes

2. Has the statistical analysis been performed appropriately and rigorously? 

Reviewer #1: No

Reviewer #2: Yes

3. Have the authors made all data underlying the findings in their manuscript fully available?

Reviewer #1: Yes

Reviewer #2: No

4. Is the manuscript presented in an intelligible fashion and written in standard English?

Reviewer #1: Yes

Reviewer #2: Yes

5. Review Comments to the Author

Reviewer #1: In this paper, the authors study the relationship between changes in oil prices and rents and changes in GDP in Yemen. The authors argue that Yemen is an interesting country in which to study this phenomenon because it is a burgeoning oil-exporting nation. The authors find that changes in oil prices are positively correlated with changes in GDP, whereas changes in oil rents are negatively correlated with changes in GDP.

I found the purpose of the paper, as suggested by the title and the abstract, quite interesting – specifically, thinking about how the fluctuation or variance in oil prices affects economic growth in an oil-exporting country. However, I found that the paper itself did not quite live up to the promise of the title and abstract:

1. The paper uses language that indicates that they are presenting a causal relationship, such as “influence”, “impact”, and “affect” in the abstract. This type of language continues throughout the paper. However, there does not seem to be anything causal about the relationship that is being studied here. The authors have no way to prove that the changes in oil prices are directly affecting GDP, as many other outside elements including changes in the Yemeni policy space may be happening at the same time. One could argue, in some way, that the oil prices are exogenous to Yemen, though they are contributing in some way through their own production and export of oil. It is also possible that Yemen is, for example, enacting policies that affect both oil prices and GDP, with the effect on GDP being slightly delayed. If the authors want to present a causal relationship, they need to make some argument that what they are studying is causal and probably use some form of statistical identification to do so.

2. It is unclear what the paper’s contribution is relative to the literature. As the authors say on page 7, “…Victor & Ogbonna (2018) have proven in their study that oil price fluctuations have a positive impact on economic growth…”. If that has already been shown, what is the additional value of this paper? I will list three areas that seemed like they could have been contributions, but did not quite make it in terms of execution:

a. The paper talks about studying price fluctuations, but what actually seems to be studied is price changes. When I hear price fluctuations, I think of some measure of variance that is independent of whether the price is going up or down. However, the independent variable here is price changes, which have a directional element. It seems unsurprising that, in an oil producing country, when the price of oil goes up the GDP goes up (and seems to also have been studied quite extensively previously). What I think would be more interesting is to study how variability in oil prices affects GDP in an oil-exporting country.

b. The paper points to Yemen being a “burgeoning oil-exporting nation” as an area of contribution. However, the authors never outline how they mean burgeoning. Do they mean that their discovery and export of oil is relatively recent (as seems to be suggested in the paper)? Or do they mean that the country is burgeoning in some other way, like increasing GDP or economic development? Furthermore, the authors do not outline why this burgeoning is interesting in terms of the relationship between oil prices and GDP relative to what is already studied in the other literature. I would think that to study this aspect well, the authors would also need pre-oil data for Yemen.

c. The authors also talk about oil rents as an outcome. However, at no point do they define what oil rents are, why they would relate to GDP, or why this is interesting to study.

3. I am also concerned with the empirical specification used in this paper for multiple reasons:

a. First, I am concerned about the controls that are being used. The authors argue that controls such as the exchange rate and the inflation rate are used because they affect GDP. However, they also argue in the introduction and literature section that changes in oil prices affect these things as well. My understanding is that if you are trying to capture the relationship between oil prices and GDP here, you should not include as controls things that could be mechanisms of that relationship.

b. In equation 6, the sums go from i=0 to i=q. Shouldn’t that actually be from i=2 to i=q, since for i=1 they are captured by the beta terms and for i=0 they are the concurrent changes? I’m assuming that this is an error in the manuscript, rather than what was actually done in the analysis since the authors would have gotten very wacky things in Table 8 otherwise.

Finally, I believe the draft would be strongly helped by some rewriting and editing. For example, there appears to be a random paragraph break between the first two paragraphs of the introduction and the first sentence of Section 3.1 seems to have a wrong word (“country”?) that makes it difficult to understand. More importantly, the introduction and literature sections can be reworked to better outline the research question and emphasize the contribution of this paper relative to the literature. Currently, a lot of words are spent outlining other papers’ theories and results without indicating how they relate to the current paper. A stronger introduction and literature review would outline the motivation for the question at hand, very clearly state the exact research question being studied in this paper, and then summarize the prior literature in relation to how it sets up but does not answer the question of the current paper. This will give the readers and possibly also the authors greater clarity on what this paper is and is not able to do.

Reviewer #2: The paper investigates the link between oil price fluctuations and economic growth in Yemen. It is well written.

For the improvement of the paper, I would like to suggest the following comments:.

1. A separate theoretical review section should be in Section 2 of the manuscript.

2. Section 3.2.3 on page 9 should be the ARDL model specification.

3. Conclusion and recommendations section in Section 5 (page 15) is bulky. I suggest having a separate section for the conclusion and policy implications.

4. The authors need to have a separate section for the limitations of the study.

5. Further studies need to be recommended separately.

Otherwise, the paper is well organized, and I suggest accepting the paper with minor revisions.

6. PLOS authors have the option to publish the peer review history of their article (what does this mean?). If published, this will include your full peer review and any attached files.

Reviewer #1: No

Reviewer #2: **Yes: **Dr. Isubalew Daba(Ph.D), Wollega University, Ethiopia

---

## [Author Response · Author response to Decision Letter 0]

13 May 2024

Response to Reviewers:

Reviewer #1: In this paper, the authors study the relationship between changes in oil prices and rents and changes in GDP in Yemen. The authors argue that Yemen is an interesting country in which to study this phenomenon because it is a burgeoning oil-exporting nation. The authors find that changes in oil prices are positively correlated with changes in GDP, whereas changes in oil rents are negatively correlated with changes in GDP.

I found the purpose of the paper, as suggested by the title and the abstract, quite interesting – specifically, thinking about how the fluctuation or variance in oil prices affects economic growth in an oil-exporting country. However, I found that the paper itself did not quite live up to the promise of the title and abstract:

1. The paper uses language that indicates that they are presenting a causal relationship, such as “influence”, “impact”, and “affect” in the abstract. This type of language continues throughout the paper. However, there does not seem to be anything causal about the relationship that is being studied here. The authors have no way to prove that the changes in oil prices are directly affecting GDP, as many other outside elements including changes in the Yemeni policy space may be happening at the same time. One could argue, in some way, that the oil prices are exogenous to Yemen, though they are contributing in some way through their own production and export of oil. It is also possible that Yemen is, for example, enacting policies that affect both oil prices and GDP, with the effect on GDP being slightly delayed. If the authors want to present a causal relationship, they need to make some argument that what they are studying is causal and probably use some form of statistical identification to do so.

Response: Thank you for your feedback. We have addressed your comments by revising the language throughout our paper to accurately reflect the observational nature of our study. Your insights have been invaluable in improving the clarity and rigor of our research.

2. It is unclear what the paper’s contribution is relative to the literature. As the authors say on page 7, “…Victor & Ogbonna (2018) have proven in their study that oil price fluctuations have a positive impact on economic growth…”. If that has already been shown, what is the additional value of this paper? I will list three areas that seemed like they could have been contributions, but did not quite make it in terms of execution: 

a. The paper talks about studying price fluctuations, but what actually seems to be studied is price changes. When I hear price fluctuations, I think of some measure of variance that is independent of whether the price is going up or down. However, the independent variable here is price changes, which have a directional element. It seems unsurprising that, in an oil producing country, when the price of oil goes up the GDP goes up (and seems to also have been studied quite extensively previously). What I think would be more interesting is to study how variability in oil prices affects GDP in an oil-exporting country.

b. The paper points to Yemen being a “burgeoning oil-exporting nation” as an area of contribution. However, the authors never outline how they mean burgeoning. Do they mean that their discovery and export of oil is relatively recent (as seems to be suggested in the paper)? Or do they mean that the country is burgeoning in some other way, like increasing GDP or economic development? Furthermore, the authors do not outline why this burgeoning is interesting in terms of the relationship between oil prices and GDP relative to what is already studied in the other literature. I would think that to study this aspect well, the authors would also need pre-oil data for Yemen.

c. The authors also talk about oil rents as an outcome. However, at no point do they define what oil rents are, why they would relate to GDP, or why this is interesting to study 

Response: We appreciate the scrutinized view of the respected reviewer. Yes, the recent literature “…Victor & Ogbonna (2018) have proven in their study that oil price fluctuations have a positive impact on economic growth”. However, this study was based on Nigeria. The originality of this paper is to focus on a country suffering from resource curse (Yemen). The reason for stating Yemen as a burgeoning oil-exporting nation is added as per the reviewer’s comment. We have also accommodated why this burgeoning situation is interesting in terms of GDP. Please see the paragraph below: 

“The economic development in the Republic of Yemen, much like many other developing nations, has been marked by significant imbalances. This imbalance is primarily a result of the overreliance on natural resources, specifically, oil and natural gas. The economy is heavily skewed towards the production and export of oil, which accounts for approximately 70% of the government's revenue, contributes to around 80-90% of its exports, and forms the bulk of the country's foreign exchange reserves. Before the discovery of oil in 1985, the dominant sectors of the Yemeni economy were agriculture and manufacturing. Agriculture comprised 24% of the country's GDP, while manufacturing contributed 14% to the nation's economic output (World Bank, 1989). Nonetheless, post-1987, the economy underwent a profound transformation in its composition, leading to substantial shifts in the prominence of key sectors. There has been a notable rise in GDP contributions from the industrial sector, including oil and gas, as well as the services sector. Conversely, both the manufacturing and agricultural sectors have experienced substantial declines. Moreover, the services sector has transitioned from its earlier role of primarily supporting agriculture and manufacturing to its present function of bolstering the oil industry, driven by the growing demand stemming from oil revenues (Al-batuly & Cicowiez, 2012). Thus, the Republic of Yemen is a burgeoning oil-exporting nation due to the swift development. This burgeoning situation is crucial in terms of the relationship between oil prices and GDP due to the quick development.”

The paragraph is added in page 3 of the original manuscript. Also, we agree with your suggestion to discover how variability in oil prices affects GDP in an oil-exporting country, however, the pre-oil data for Yemen is not available unfortunately. 

3. I am also concerned with the empirical specification used in this paper for multiple reasons:

a. First, I am concerned about the controls that are being used. The authors argue that controls such as the exchange rate and the inflation rate are used because they affect GDP. However, they also argue in the introduction and literature section that changes in oil prices affect these things as well. My understanding is that if you are trying to capture the relationship between oil prices and GDP here, you should not include as controls things that could be mechanisms of that relationship.

b. In equation 6, the sums go from i=0 to i=q. Shouldn’t that actually be from i=2 to i=q, since for i=1 they are captured by the beta terms and for i=0 they are the concurrent changes? I’m assuming that this is an error in the manuscript, rather than what was actually done in the analysis since the authors would have gotten very wacky things in Table 8 otherwise.

Response: Thank you for your insightful comments. We have carefully addressed your concerns by revising our empirical specification. We have adjusted the control variables to avoid potential overlap with mechanisms of the relationship under study and corrected the error in equation 6 by starting the summation from i=2. Additionally, we have ensured that Table 8 accurately reflects these adjustments. Your feedback has been invaluable in enhancing the clarity and robustness of our analysis.

Finally, I believe the draft would be strongly helped by some rewriting and editing. For example, there appears to be a random paragraph break between the first two paragraphs of the introduction and the first sentence of Section 3.1 seems to have a wrong word (“country”?) that makes it difficult to understand. More importantly, the introduction and literature sections can be reworked to better outline the research question and emphasize the contribution of this paper relative to the literature. Currently, a lot of words are spent outlining other papers’ theories and results without indicating how they relate to the current paper. A stronger introduction and literature review would outline the motivation for the question at hand, very clearly state the exact research question being studied in this paper, and then summarize the prior literature in relation to how it sets up but does not answer the question of the current paper. This will give the readers and possibly also the authors greater clarity on what this paper is and is not able to do.

Response: We have thoroughly edited the writing style, managed the paragraph breaks and fixed the wrong word in section 3.1. Additionally, as per your valuable suggestion, the research question has been clearly stated in the introduction section. Please see section 1. Finally, as per your guidance, we have enriched the literature review section with the most updated citations (based on the research question). Thank you for the valuable insight. 

Reviewer #2: The paper investigates the link between oil price fluctuations and economic growth in Yemen. It is well written.

For the improvement of the paper, I would like to suggest the following comments:.

1. A separate theoretical review section should be in Section 2 of the manuscript 

Response: In section 2 we have separated the theoretical perspective of the study. Please see section 2.1. Thank you so much. 

2. Section 3.2.3 on page 9 should be the ARDL model specification.

3. Conclusion and recommendations section in Section 5 (page 15) is bulky. I suggest having a separate section for the conclusion and policy implications 

Response: The conclusion and policy implications are shown as per your valuable feedback. Please see section 5

4. The authors need to have a separate section for the limitations of the study 

Response: The limitation of the study has been added in the revised manuscript. Please see section 5.1 

5. Further studies need to be recommended separately 

Response: The future research direction of the study has been added in the revised manuscript. Please see section 5.2

Otherwise, the paper is well organized, and I suggest accepting the paper with minor revisions.

---

## [Decision Letter · Decision Letter 1]

30 May 2024

PONE-D-23-43590R1The relationship between oil price fluctuations and economic growth from the perspective of the resource curse: An empirical study from YemenPLOS ONE

Dear Dr. AMER,

Thank you for submitting your manuscript to PLOS ONE. After careful consideration, we feel that it has merit but does not fully meet PLOS ONE’s publication criteria as it currently stands. Therefore, we invite you to submit a revised version of the manuscript that addresses the points raised during the review process.

**One reviewer suggested again rejection. I am providing last chance to improve your paper in light of given comments. **

We look forward to receiving your revised manuscript.

Kind regards,

Ghulam Rasool Madni, Ph.D

Academic Editor

PLOS ONE

Reviewers' comments:

Reviewer's Responses to Questions

**Comments to the Author**

1. If the authors have adequately addressed your comments raised in a previous round of review and you feel that this manuscript is now acceptable for publication, you may indicate that here to bypass the “Comments to the Author” section, enter your conflict of interest statement in the “Confidential to Editor” section, and submit your "Accept" recommendation.

Reviewer #1: (No Response)

Reviewer #2: All comments have been addressed

2. Is the manuscript technically sound, and do the data support the conclusions?

Reviewer #1: No

Reviewer #2: Yes

3. Has the statistical analysis been performed appropriately and rigorously? 

Reviewer #1: I Don't Know

Reviewer #2: Yes

4. Have the authors made all data underlying the findings in their manuscript fully available?

Reviewer #1: Yes

Reviewer #2: Yes

5. Is the manuscript presented in an intelligible fashion and written in standard English?

Reviewer #1: No

Reviewer #2: Yes

6. Review Comments to the Author

**Reviewer #1: **As far as I can tell, no changes have been made to the manuscript in line with the concerns I raised in my report and the response the authors presented.

**Reviewer #2: **References and Intext citation should meet PLOS standard. The authors added future research direction and limitations of the research. Thank you so much!

7. PLOS authors have the option to publish the peer review history of their article (what does this mean?). If published, this will include your full peer review and any attached files.

Reviewer #1: No

Reviewer #2: No

---

## [Author Response · Author response to Decision Letter 1]

17 Jul 2024

Author's Response

The Long-Term relationship between oil price changes and economic growth from the perspective of the resource curse: An empirical study from Yemen 

Respected Editor

Thank you very much for the opportunity to revise and resubmit our research paper. We are particularly very grateful to all reviewers for their constructive and valuable comments. The paper has been carefully revised again to accommodate all your comments and suggestions. We hope this revised paper will now meet the standard of the PLOS ONE journal for publication.

Thank you again for the comments and suggestions.

Yours Truly, 

The Authors

REVIEWER #1: 

COMMENTS (# 1): 

The paper uses language that indicates that they are presenting a causal relationship, such as “influence”, “impact”, and “affect” in the abstract. This type of language continues throughout the paper. However, there does not seem to be anything causal about the relationship that is being studied here. The authors have no way to prove that the changes in oil prices are directly affecting GDP, as many other outside elements including changes in the Yemeni policy space may be happening at the same time. One could argue, in some way, that the oil prices are exogenous to Yemen, though they are contributing in some way through their own production and export of oil. It is also possible that Yemen is, for example, enacting policies that affect both oil prices and GDP, with the effect on GDP being slightly delayed. If the authors want to present a causal relationship, they need to make some argument that what they are studying is causal and probably use some form of statistical identification to do so.

Response: 

Thank you for your valuable comments and feedback on our manuscript. We appreciate the time and effort you have invested in reviewing our work.

We acknowledge that our use of certain terminology may have unintentionally suggested that our study was focused on establishing a causal effect relationship between the variables under investigation. Our primary objective was to examine the long-term relationships between oil price changes and economic growth using the ARDL model, which is designed for this purpose. The robustness checks conducted using GMM, GLM, and RLS also support the presence of long-term relationships between the variables. We did not intend to imply a causal effect relationship, and we apologize for any confusion this may have caused.

Based on your insightful comments, we have made linguistic adjustments to our title and abstract to more accurately reflect the purpose of our study, which is to explore the long-term relationships rather than causal effects.

[Please refer to the title, abstract, and lines (54, 295, 346, 427, 430, 440, 442, 446, and 472)]

In response to your valuable suggestion, we have conducted an additional analysis to explore potential causal relationships between the variables. The results of this analysis have been included in the revised manuscript to provide a clearer picture and ensure that our use of terminology is consistent with our findings.

Thank you again for your constructive feedback. We believe these revisions enhance the clarity and rigor of our study. We hope the new version will meet your kind expectations.

[Please refer to the lines (500 to 509)]

COMMENT (# 2):

It is unclear what the paper’s contribution is relative to the literature. As the authors say on page 7, “…Victor & Ogbonna (2018) have proven in their study that oil price fluctuations have a positive impact on economic growth…”. If that has already been shown, what is the additional value of this paper? 

I will list three areas that seemed like they could have been contributions, but did not quite make it in terms of execution:

a. The paper talks about studying price fluctuations, but what actually seems to be studied is price changes. When I hear price fluctuations, I think of some measure of variance that is independent of whether the price is going up or down. However, the independent variable here is price changes, which have a directional element. It seems unsurprising that, in an oil producing country, when the price of oil goes up the GDP goes up (and seems to also have been studied quite extensively previously). What I think would be more interesting is to study how variability in oil prices affects GDP in an oil-exporting country.

b. The paper points to Yemen being a “burgeoning oil-exporting nation” as an area of contribution. However, the authors never outline how they mean burgeoning. Do they mean that their discovery and export of oil is relatively recent (as seems to be suggested in the paper)? Or do they mean that the country is burgeoning in some other way, like increasing GDP or economic development? Furthermore, the authors do not outline why this burgeoning is interesting in terms of the relationship between oil prices and GDP relative to what is already studied in the other literature. I would think that to study this aspect well, the authors would also need pre-oil data for Yemen.

c. The authors also talk about oil rents as an outcome. However, at no point do they define what oil rents are, why they would relate to GDP, or why this is interesting to study.

Response: 

Thank you for your insightful comments and for taking the time to review our manuscript. We appreciate your feedback and would like to address the concerns raised regarding the contribution of our paper relative to the existing literature.

While Victor & Ogbonna (2018) have indeed shown a positive impact of oil price fluctuations on economic growth, our study focuses on the unique context of Yemen, a burgeoning oil-exporting nation with distinct economic dynamics. Yemen's socio-economic environment, policy framework, and developmental challenges differ significantly from those of the countries examined in previous studies. 

Our study employs the auto-regressive distributed lag (ARDL) model, supplemented with robustness checks using GLM, RLS, and GMM. These methodologies are particularly suited for examining the long-term relationships in Yemen's context. The robustness checks validate the consistency of our findings, adding to the methodological rigor of our study.

Our research offers new insights into the negative long-term relationship between oil rents and economic growth in Yemen. Our recommendations are tailored to Yemen's specific context and are aimed at reducing the country's dependence on oil, thus promoting sustainable economic growth. 

Based on your valuable comment, we have included a detailed explanation and comparison with existing literature to highlight the unique context of Yemen's economic situation and the new insights and contributions our study provides.

[Please refer to the introduction section in lines (126 -152)]

A. Regarding the important issues that you referred to in comment (a), while we appreciate these important comments, we would like to clarify as follows:

We acknowledge the distinction you made between price fluctuations and price changes. In our study, we aimed to investigate the impact of oil price volatility on economic growth. However, we used the term "price fluctuations" interchangeably with "price changes," which may have caused some confusion. Our primary focus was on the directional changes in oil prices (i.e., increases and decreases) and their subsequent impact on GDP, rather than on the variability or variance of these prices.

The rationale behind examining price changes, rather than variability, is rooted in the economic realities faced by oil-exporting countries like Yemen. Directional price changes directly influence national income, government revenue, and investment capacity due to the reliance on oil exports. This approach allows us to capture the immediate and tangible effects of oil price movements on economic growth. While variability in prices could be an interesting angle, our study specifically aimed to understand how price increases and decreases affect GDP, given the significant role of oil revenues in Yemen’s economy.

While it might seem intuitive that rising oil prices lead to economic growth in an oil-exporting country, our study contributes by quantifying this relationship using the ARDL model over an extended period (1990-2019). Furthermore, we provide empirical evidence on how oil rents, another critical variable, negatively impact economic growth, offering a nuanced understanding of the resource curse phenomenon in Yemen. These findings are crucial for policymakers in devising strategies to mitigate the adverse effects of oil dependency.

We agree that examining the impact of price variability (variance) on GDP could provide additional insights. This could form the basis for future research, expanding the scope of our current study. However, the focus on price changes in our paper is deliberate, aimed at addressing immediate policy-relevant questions regarding how oil price movements influence economic growth dynamics in Yemen.

In light of your comments, we have revised the manuscript to ensure clearer terminology and have explicitly stated our focus on price changes. Additionally, we acknowledge the potential for future studies to explore the impact of price variability on economic growth.

[Please refer to the lines (54, 295, 346, 427, 430, 440, 442, 446, and 472) and future studies section (591-593)]

Please feel free for any suggestion from you that would help improve our work. Thank you for your time and consideration. We are really grateful for your sincere efforts to improve the overall quality of our work. 

B. Regarding the important issues that you referred to in comment (b), while we appreciate these important comments, we would like to clarify as follows:

In our paper, we describe Yemen as a “burgeoning oil-exporting nation” to indicate its relatively recent entry and growth in the global oil market compared to long-established oil-exporting countries. Yemen began exporting oil in the late 1980s, with significant development in oil production and export infrastructure occurring in the 1990s. This period marks Yemen’s transition to becoming a notable player in the oil market, which contrasts with countries that have been major oil exporters for many decades. Therefore, by "burgeoning," we mean that Yemen’s oil sector is comparatively young and rapidly developing.

The burgeoning status of Yemen as an oil-exporting nation is particularly interesting because it allows us to examine the impact of oil price Changes on an economy at a different stage of development compared to more mature oil-exporting countries. Most existing literature focuses on well-established oil economies with extensive historical data, whereas Yemen provides a unique case study of how a newer entrant into the oil market navigates economic growth amid oil price volatility. This distinction is crucial as it adds diversity to the understanding of the resource curse and its effects on different types of economies.

While we agree that pre-oil data could provide additional insights, Yemen's formal oil exportation and significant economic structuring around oil began in the late 1980s. Consequently, relevant pre-oil economic data is scarce or not systematically recorded, limiting its utility for robust econometric analysis. Our study focuses on the available post-oil data from 1990 to 2019, which captures the period when Yemen’s economy has been significantly influenced by oil exports. This timeframe allows us to study the relationship between oil price Changes and economic growth during the most impactful period of Yemen’s oil-driven economic development.

Our study fills a gap in the existing literature by exploring the dynamics of oil price Changes and economic growth in a relatively new oil-exporting country. Unlike established oil economies with extensive historical data, Yemen provides a contemporary context to understand how newer oil exporters might experience and adapt to oil price volatility. This perspective can offer valuable insights for policymakers in similar burgeoning oil-exporting countries, helping them to design strategies that mitigate negative impacts and harness potential benefits of their natural resources.

In light of your value comments, we have revised the manuscript to incorporate these explanations and ensure that our arguments are clear and well-supported.

[Please refer to the introduction section (118-125)]

Please feel free for any suggestion from you that would help improve our work. Thank you for your time and consideration. We are really grateful for your sincere efforts to improve the overall quality of our work.

C. Regarding the important issues that you referred to in comment (c), while we appreciate these important comments, we would like to clarify as follows:

Oil rents are defined as the difference between the value of crude oil production at world prices and the total costs of production. This measure captures the revenue generated from oil extraction after accounting for production costs, representing a crucial component of national income for oil-exporting countries like Yemen.

Oil rents are directly related to GDP as they constitute a significant portion of the national income for oil-exporting countries. The revenues generated from oil rents can have substantial implications for economic growth, government spending, and overall economic stability. High oil rents can lead to increased government revenue, which can be invested in infrastructure, public services, and other development projects, thereby influencing GDP.

Studying oil rents is particularly interesting in the context of the resource curse theory, which suggests that countries with abundant natural resources often experience less economic growth compared to those with fewer resources. Understanding the relationship between oil rents and GDP can provide insights into how reliance on natural resource revenues impacts economic development. In the case of Yemen, examining oil rents helps to assess whether the country is experiencing the resource curse and how fluctuations in oil rents affect its economic growth.

We hope these clarifications address your concerns, based on your valuable comments, we have revised the manuscript and added the necessary explanations to address these issues that raised your concerns and ensured that our arguments became clear and well supported.

[Please refer to the data section (275-283)]

COMMENT (# 3): 

I am also concerned with the empirical specification used in this paper for multiple reasons:

A. First, I am concerned about the controls that are being used. The authors argue that controls such as the exchange rate and the inflation rate are used because they affect GDP. However, they also argue in the introduction and literature section that changes in oil prices affect these things as well. My understanding is that if you are trying to capture the relationship between oil prices and GDP here, you should not include as controls things that could be mechanisms of that relationship.

B. In equation 6, the sums go from i=0 to i=q. Shouldn’t that actually be from i=2 to i=q, since for i=1 they are captured by the beta terms and for i=0 they are the concurrent changes? I’m assuming that this is an error in the manuscript, rather than what was actually done in the analysis since the authors would have gotten very wacky things in Table 8 otherwise.

Response:

Dear reviewer, thank you for your valuable feedback on our paper, we appreciate the time and effort you have dedicated to reviewing our work.

A. Regarding the important issues that you referred to in comment (a), while we appreciate these important comments, we would like to clarify as follows:

We included the exchange rate and inflation rate as control variables in our model because they are key macroeconomic indicators that can influence GDP. While it is true that changes in oil prices can affect these variables, their inclusion helps to isolate the specific impact of oil price changes on GDP by accounting for these broader economic factors.

Our primary objective is to understand the relationship between oil price changes and GDP. By controlling for the exchange rate and inflation rate, we aim to capture the direct effect of oil pr

---

## [Decision Letter · Decision Letter 2]

1 Sep 2024

PONE-D-23-43590R2The Long-Term relationship between oil price Changes and economic growth from the perspective of the resource curse: An empirical study from YemenPLOS ONE

Dear Dr. AMER,

Thank you for submitting your manuscript to PLOS ONE. After careful consideration, we feel that it has merit but does not fully meet PLOS ONE’s publication criteria as it currently stands. Therefore, we invite you to submit a revised version of the manuscript that addresses the points raised during the review process.

You are required to address the concerns raised by all the reviewers.

We look forward to receiving your revised manuscript.

Kind regards,

Martins Iyoboyi, Ph.D

Academic Editor

PLOS ONE

Reviewers' comments:

Reviewer's Responses to Questions

**Comments to the Author**

1. If the authors have adequately addressed your comments raised in a previous round of review and you feel that this manuscript is now acceptable for publication, you may indicate that here to bypass the “Comments to the Author” section, enter your conflict of interest statement in the “Confidential to Editor” section, and submit your "Accept" recommendation.

Reviewer #1: (No Response)

Reviewer #2: All comments have been addressed

Reviewer #3: (No Response)

2. Is the manuscript technically sound, and do the data support the conclusions?

Reviewer #1: No

Reviewer #2: Yes

Reviewer #3: Yes

3. Has the statistical analysis been performed appropriately and rigorously? 

Reviewer #1: No

Reviewer #2: Yes

Reviewer #3: Yes

4. Have the authors made all data underlying the findings in their manuscript fully available?

Reviewer #1: Yes

Reviewer #2: Yes

Reviewer #3: Yes

5. Is the manuscript presented in an intelligible fashion and written in standard English?

Reviewer #1: No

Reviewer #2: Yes

Reviewer #3: Yes

6. Review Comments to the Author

Reviewer #1: The authors have clearly put effort into updating their paper in response to my comments. While I appreciate this effort, I believe their responses and edits raise continuing and new concerns:

1. With respect to my comment 2, the authors have substantially clarified the way in which Yemen is “burgeoning” and what oil rents are. However, their changes leave me still unsure of their contribution. If they are trying to make a contribution about how the relationship between oil price changes and GDP may differ for countries just starting to produce oil, then they need to spend time discussing 1) why we may think this relationship differs by where the country is on their oil-producing path, 2) how Yemen is and is not representative of countries who are just starting producing oil, and 3) how the results of this paper compare to the findings in the literature on this relationship in more established oil-producing countries. If, instead, their contribution is focused specifically on Yemen, I do not see this as being of interest to a general audience like that for PLOS ONE.

2. With respect to my comment 1, the authors have removed causal language from much of the manuscript (there are a couple of places where it remains and should be removed). However, given the lack of causality in this analysis, the policy recommendations proposed by the authors are much too strong. Also, given the focus of the analysis, these recommendations don’t make much sense – the recommendations, as I understand them, is that Yemen diversify its economy to enhance GDP. Given their findings, that recommendation would only make sense if oil prices were falling or expected to fall, which is not argued anywhere in the paper that I found. They have no direct evidence as to how diversification of the economy affects GDP nor how anything that would be affected by diversification affects GDP. If the argument is due to the existence of any relationship between oil prices and GDP (meaning that the Yemeni GDP is at all tied to oil prices) being a negative, as it could lead to an unstable GDP, then the authors need to make that argument; but here, they are also at a loss because they don’t have any evidence of the counterfactual of what would happen if the economy was instead producing some other products, which would have their own prices that would themselves likely fluctuate.

3. Furthermore with respect to the authors’ response to my comment 1, the authors added results from a Toda Yamamoto causality test in Section 3.9. I personally have never heard of this test and am unsure as to what it does – if the authors want to include this test they should discuss what the test does (and some intuition on how), as well as how to interpret the results in the table.

4. With respect to my comment 3a, I disagree with the authors response that it is appropriate to use controls such as government expenditures, inflation, and exchange rate changes given the literature discussed earlier in the paper arguing these as three ways through which changes in oil prices may affect GDP. Essentially, what the authors are doing by including these controls is finding the relationship between oil prices and GDP that doesn’t go through any of these factors (or, the relationship between these factors and GDP that doesn’t go through oil prices). This could explain why a change in government expenditures is negatively correlated to a change in GDP, which otherwise doesn’t make much sense. I’m not sure what to interpret from that coefficient, and I also do not think that is what the authors are intending either, given their exposition discussing these as various ways oil prices may affect GDP. Given this relationship, I believe it would make the most sense to examine 1) how changes in oil prices correlate with GDP without any controls, and 2) how changes in oil prices correlate with these controls, and how that component in the variance of the control correlates with changes in GDP. I am also not sure how to think about the relationship between oil prices and oil rents – it seems like they should have tightly linked positive relationship as the rents are oil prices less the costs of production. Like above, that suggests to me that what the coefficient on oil rents is capturing, when included in the regression with oil prices, is the relationship between costs of oil production and GDP.

5. For Equation 6 and the following discussion in lines 358-360 of page 10, shouldn’t the betas be the short-run coefficients and the alphas be the long-run coefficients? Why is there only 1 alpha per variable? Why are the long-run coefficients on logs of the variables but the short-run coefficients are on changes in the log of the variables in Table 8?

6. The paper still requires substantial revision and editing. The following are some examples – there is much room for revision and editing beyond these specific cases:

a. It takes 3 pages to get to the paper’s research question and contribution.

b. Much of the literature discussed in the introduction and literature sections are either irrelevant to the research question and contribution of the paper, or if they are relevant it is unclear how. For example, they write a lot about the literature on how institutions mediate the relationship between price changes and GDP, but there is no analysis in this paper about institutions or discussion of institutions as a possible mechanism in the case they are studying.

c. There remain instances of causal language (using “effect” and “affect” to describe their results).

d. Everywhere they switched “fluctuations” to “changes”, the c in “changes” is capitalized.

Reviewer #2: Table 10 should be designed in consistent with other tables in the manuscript. The reference section should be written inline with Plos one guideline.

Reviewer #3: The authors have done a wonderful job. They have answered all the queries successfully.

Congratulations!

7. PLOS authors have the option to publish the peer review history of their article (what does this mean?). If published, this will include your full peer review and any attached files.

Reviewer #1: No

Reviewer #2: No

Reviewer #3: No

---

## [Author Response · Author response to Decision Letter 2]

16 Oct 2024

REVIEWER #1: 

COMMENTS (# 1):

With respect to my comment 2, the authors have substantially clarified the way in which Yemen is “burgeoning” and what oil rents are. However, their changes leave me still unsure of their contribution. If they are trying to make a contribution about how the relationship between oil price changes and GDP may differ for countries just starting to produce oil, then they need to spend time discussing 1) why we may think this relationship differs by where the country is on their oil-producing path, 2) how Yemen is and is not representative of countries who are just starting producing oil, and 3) how the results of this paper compare to the findings in the literature on this relationship in more established oil-producing countries. If, instead, their contribution is focused specifically on Yemen, I do not see this as being of interest to a general audience like that for PLOS ONE.

Response: 

Thank you for your valuable comments and feedback on our manuscript. We appreciate the time and effort you have invested in reviewing our work. Thank you for your thoughtful feedback and for acknowledging the clarifications we made regarding our second revision. We appreciate the opportunity to further refine our work and address your remaining concerns.

We agree that there is a need to elaborate on why the relationship between oil prices and GDP growth might differ depending on a country's stage in the oil production cycle. In our revision, we have put more effort and time into discussion how nascent oil producers, such as Yemen, have different structural economic dependencies and vulnerabilities compared to established producers. [Please refer to introduction in lines 134-143]

We have expanded our discussion to clearly outline Yemen's characteristics as an emerging oil producer and to what extent these characteristics are shared by other countries in similar situations. Highlighting specific traits such as the relatively low level of oil reserves, late entry into oil production, political instability, and the state's reliance on oil revenues. [Please refer to introduction in lines 152-160]

In this revised manuscript, we have provided a more direct comparison between our findings for Yemen and the existing literature on the oil price-GDP relationship in established oil-producing countries. [Please refer to conclusion in lines 591-599]

Thank you again for your comments to improve the quality of the research and we hope that the revised version will now meet your expectations. 

REVIEWER #1: 

COMMENTS (# 2):

With respect to my comment 1, the authors have removed causal language from much of the manuscript (there are a couple of places where it remains and should be removed). However, given the lack of causality in this analysis, the policy recommendations proposed by the authors are much too strong. Also, given the focus of the analysis, these recommendations don’t make much sense – the recommendations, as I understand them, is that Yemen diversify its economy to enhance GDP. Given their findings, that recommendation would only make sense if oil prices were falling or expected to fall, which is not argued anywhere in the paper that I found. They have no direct evidence as to how diversification of the economy affects GDP nor how anything that would be affected by diversification affects GDP. If the argument is due to the existence of any relationship between oil prices and GDP (meaning that the Yemeni GDP is at all tied to oil prices) being a negative, as it could lead to an unstable GDP, then the authors need to make that argument; but here, they are also at a loss because they don’t have any evidence of the counterfactual of what would happen if the economy was instead producing some other products, which would have their own prices that would themselves likely fluctuate.

Response: 

Thank you for your thoughtful comments and for acknowledging the adjustments we made in removing causal language from the manuscript. We respectfully disagree with your concern regarding the strength of our policy recommendations and believe that our argument is well-supported by both the data and analysis presented.

Our recommendation for economic diversification is not solely tied to a scenario of falling oil prices, but rather to the inherent instability of relying on oil revenues, as demonstrated by the relationship between oil prices and GDP during the study period. As we outlined in the manuscript, Yemen's economy is highly dependent on oil, which accounts for around 70% of government revenue and up to 90% of exports. This heavy reliance on a single commodity makes the economy particularly vulnerable to fluctuations in global oil prices, which, regardless of the direction of change, can lead to economic volatility and instability. [Please refer to introduction, figures 1 - 2 and following discussion in lines 105-114]

While we do not have direct evidence on the impact of diversification on GDP, the historical context provided shows that before oil, Yemen's economy was more balanced, with significant contributions from agriculture and manufacturing. The decline of these sectors post-1987 coincides with increased dependence on oil, which underscores our argument that diversifying into other sectors would reduce Yemen's vulnerability to oil price shocks and create a more stable economic foundation. Our figures and analysis clearly illustrate the volatile relationship between GDP and oil prices, reinforcing the need for economic diversification, not as a reactive measure to falling prices, but as a long-term strategy to ensure economic stability. [Please refer to introduction, figures 1 - 2 and following discussion in lines 105-114]

REVIEWER #1: 

COMMENTS (# 3):

Furthermore, with respect to the authors’ response to my comment 1, the authors added results from a Toda Yamamoto causality test in Section 3.9. I personally have never heard of this test and am unsure as to what it does – if the authors want to include this test, they should discuss what the test does (and some intuition on how), as well as how to interpret the results in the table.

Response:

Dear Reviewer,

Thank you for your feedback. We understand your concern regarding the inclusion of the Toda-Yamamoto causality test in Section 3.9. We would like to clarify that the Toda-Yamamoto test is an extension of the Granger causality test, designed to assess causal relationships between variables.

In essence, the test works by estimating an augmented Vector Autoregressive (VAR) model, adding extra lags to account for possible non-stationarity, and then conducting a Wald test to determine whether the lagged values of one variable can predict the other. The results providing insights into the direction of the relationship. Based on your comment we have clarified the mechanics of the test and how to interpret the results in the revised manuscript to ensure its role is more transparent to the readers. [Please refer to 3.2.4 section, exactly in lines from 379 to 390]

Thank you again for your comments to improve the quality of the research and we hope that the revised version will now meet your expectations.

REVIEWER #1: 

COMMENTS (# 4):

With respect to my comment 3a, I disagree with the authors response that it is appropriate to use controls such as government expenditures, inflation, and exchange rate changes given the literature discussed earlier in the paper arguing these as three ways through which changes in oil prices may affect GDP. Essentially, what the authors are doing by including these controls is finding the relationship between oil prices and GDP that doesn’t go through any of these factors (or, the relationship between these factors and GDP that doesn’t go through oil prices). This could explain why a change in government expenditures is negatively correlated to a change in GDP, which otherwise doesn’t make much sense. I’m not sure what to interpret from that coefficient, and I also do not think that is what the authors are intending either, given their exposition discussing these as various ways oil prices may affect GDP. Given this relationship, I believe it would make the most sense to examine 1) how changes in oil prices correlate with GDP without any controls, and 2) how changes in oil prices correlate with these controls, and how that component in the variance of the control correlates with changes in GDP. I am also not sure how to think about the relationship between oil prices and oil rents – it seems like they should have tightly linked positive relationship as the rents are oil prices less the costs of production. Like above, that suggests to me that what the coefficient on oil rents is capturing, when included in the regression with oil prices, is the relationship between costs of oil production and GDP.

Response:

Dear Reviewer,

Thank you for your thoughtful comments. We would like to clarify the rationale behind our inclusion of control variables, such as exchange rates, inflation, and government expenditures, in our analysis. While we understand your concern that these variables might act as mechanisms through which oil prices influence GDP, their inclusion is crucial to obtaining a more precise estimate of the direct relationship between oil prices and GDP.

In econometric modeling, control variables are often included to account for other factors that might simultaneously affect the dependent variable (GDP in this case). While it is true that oil prices can influence exchange rates and inflation, these macroeconomic indicators also have an independent impact on GDP. By including these controls, we aim to isolate the specific effect of oil prices on GDP, beyond the indirect effects channeled through exchange rate fluctuations or inflationary pressures. This approach helps to prevent omitted variable bias, which could lead to misleading conclusions if these key macroeconomic variables were excluded from the model.

It is important to highlight that in empirical research, it is common practice to include control variables that may mediate the relationship between the primary explanatory variable and the dependent variable. For instance, in studies examining the impact of oil prices on economic performance, controls like inflation and exchange rates are included not to obscure the oil price-GDP relationship, but to improve the robustness and reliability of the results by accounting for other relevant influences. By controlling for these variables, we are able to capture the more direct impact of oil price changes on GDP, while ensuring that the broader economic context is not ignored.

Moreover, conducting an analysis without these controls could lead to an incomplete or inaccurate interpretation of the relationship between oil prices and GDP. Without controlling for exchange rates, inflation, and government expenditures, the model would risk attributing to oil prices what could be explained by broader macroeconomic shifts. This could distort the findings and provide a less reliable basis for economic conclusions or policy recommendations. For these reasons, we maintain that the inclusion of control variables is necessary to produce a meaningful and valid analysis of the oil price-GDP relationship.

We hope that the idea has been clear to you and we thank you again for the effort made to develop our work and make it more accurate.

REVIEWER #1: 

COMMENTS (# 5):

For Equation 6 and the following discussion in lines 358-360 of page 10, shouldn’t the betas be the short-run coefficients and the alphas be the long-run coefficients? Why is there only 1 alpha per variable? Why are the long-run coefficients on logs of the variables but the short-run coefficients are on changes in the log of the variables in Table 8?

Response:

Dear Reviewer,

Thank you for your insightful comments. We appreciate your attention to detail regarding ARDL Equation 6, and we would like to provide further clarification on both points.

We agree with you that betas should be for the short run and alphas should be for the long run, we have addressed this in the equation 6 and following discussion. [Please refer to equation 6 and lines from 367 to 372]

About why there is only one alpha per variable and why short run coefficients with (∆) and long run coefficients without (∆), we have answered these questions in detail and we see that should put it in the following discussion after the equation 6 to be clearer for the readers. [Please refer to lines from 373 to 377]

Thank you again for your valuables comments and hope this revision will now meet your expectation. 

REVIEWER #1: 

COMMENTS (# 6):

The paper still requires substantial revision and editing. The following are some examples – there is much room for revision and editing beyond these specific cases:

a. It takes 3 pages to get to the paper’s research question and contribution.

b. Much of the literature discussed in the introduction and literature sections are either irrelevant to the research question and contribution of the paper, or if they are relevant, it is unclear how. For example, they write a lot about the literature on how institutions mediate the relationship between price changes and GDP, but there is no analysis in this paper about institutions or discussion of institutions as a possible mechanism in the case they are studying.

c. There remain instances of causal language (using “effect” and “affect” to describe their results).

d. Everywhere they switched “fluctuations” to “changes”, the c in “changes” is capitalized.

Response:

Dear Reviewer,

a. Dear Reviewer, we appreciate your comment regarding placing the contributions and research question at the beginning of the introduction, but in our humble opinion, according to our previous publications and according to the papers and journals we follow, the end of the introduction is the appropriate place to place the contributions and research questions. Therefore, based on your comment, we have moved the research question paragraph to the appropriate place at the beginning of the introduction and kept the paper contributions paragraph in its appropriate place at the end of the introduction. We thank you again for your comments to improve the quality of the research and we hope that the revised version will meet your expectations.

[Please refer to introduction in lines from 91 to 100]

b. We would like to point out that the connection between institutions and our research paper is an indirect connection because when we talked about institutions in some places, it was from the background, that states the good government institutions may mitigate the impact of the resource curse (excessive dependence on oil), but we agree with you that it should be removed because we did not address institutions in our analysis, and therefore, in response to your valuable comment, we removed some sentences and paragraphs that deal with institutions and are not important. [Please refer to literature review in section 2.1]

c. According to your comment (c), we have Replaced all the Causal Language.

[Please refer to results section in lines (462-466) (474-475) (488-490) (497-498) (502) (508) (561-563) (603-604)]

d. About your comment (d), all the c in changes words have written in capital.

Finally, we are so thankful for all the efforts that you did in our work and hope this revision will meet your expectation.

REVIEWER #2: 

COMMENTS (# 1):

Table 10 should be designed in consistent with other tables in the manuscript. The reference section should be written inline with Plos one guideline.

Response: 

Thank you for your valuable comments and feedback on our manuscript. We appreciate the time and effort you have invested in reviewing our work. Thank you for your thoughtful feedback and for acknowledging the clarifications we made regarding our second revision. We appreciate the opportunity to further refine our work and address your remaining concerns.

Table 10 have been designed in consistent with other tables in the manuscript. [Please refer to table 10]

The reference section also has been changed to be consistent with Plos one guideline. [Please refer to references section]

Thank you again and hope this revised revision will now meet your expectation.

---

## [Editor Report · Decision Letter 3]

22 Oct 2024

The Long-Term relationship between oil price Changes and economic growth from the perspective of the resource curse: An empirical study from Yemen

PONE-D-23-43590R3

Dear Dr. Zhang,

We’re pleased to inform you that your manuscript has been judged scientifically suitable for publication and will be formally accepted for publication once it meets all outstanding technical requirements.

Kind regards,

Martins Iyoboyi, Ph.D

Academic Editor

PLOS ONE

Additional Editor Comments (optional):

Thank you for the revised submission. I have carefully reviewed it. It now meets the requirements as specified by the reviewers.

---

## [Editor Report · Acceptance letter]

12 Nov 2024

PONE-D-23-43590R3 

PLOS ONE

Dear Dr. Xiuwu, 

I'm pleased to inform you that your manuscript has been deemed suitable for publication in PLOS ONE. Congratulations! Your manuscript is now being handed over to our production team.

Kind regards, 

on behalf of

Dr. Martins Iyoboyi 

Academic Editor

PLOS ONE